# Nonequilibrium pulsed heating freezes sintering of supported metal nanocatalysts

Jiawei Huang [1], Zhouyang Zhang [2] ✉, Guangren Wang[1], Jiaqi Chen[1], Yucheng Zhang[1], Jian Zhou[1], Chunxian Xing[1], Yiran Ying [3] ✉, Changshui Huang [4] ✉ & Linfeng Fei [1] ✉

Supported metal nanoparticles always experience thermal-induced sintering that severely depresses their catalytic performance and durability. Recently, ultrafast heating technology has been widely adopted in synthesizing diverse metal nanoparticles or even single atoms on supporting materials, offering enhanced anti-sintering capabilities compared to conventional heating protocols. However, the mechanisms and kinetics underlying this anti-sintering behavior during ultrafast heating processes remain poorly understood. Here, using in situ scanning transmission electron microscopy and theoretical analysis, we microscopically reveal a metastable state for supported metal nanoparticles induced by ultrafast heating pulses, which significantly mitigates the thermal-induced sintering while effectively improves the crystallization degree of metal nanoparticles together with the metal/support interfaces. Our results show that Pt nanoparticles supported on graphene flakes are thermodynamically unstable yet kinetically stable across the ultrafast heating pulses; as a consequence, the Pt/graphene interface is gradually optimized in a lattice scale, leading to remarkable sintering resistance for Pt nanoparticles. These atomic-scale insights provide thorough fundamental understandings for ultrafast heating in stabilizing metal nanoparticles and may further guide the high-throughput production of robust supported metal nanocatalysts.

The development of highly efficient and stable nanocatalysts represents one of the central objectives in current materials science, driven by their extensive applications expanding from energy transformation to chemical manufacturing fields.[1–6] Particularly, most of these state-of-the-art nanocatalysts are metal nanoparticles (NPs, ranging from single atoms to several nanometers in size) supported on inert substrates, designing for maintaining the enormous surface areas as well as the high catalytic efficiencies of metal NPs in complex catalytic environments.[7–10] However, the synthesis and deployment of supported metal nanocatalysts inevitably involve high-temperature treatments (for decomposing the precursors,[11–13] or improving the crystallization degrees of metal NPs during the synthesis,[14,15] or activating the catalytic reactions during the applications of nanocatalysts[16,17]), which usually include a sequential heating/holding/cooling program lasting for several hours. In this context, a common issue, which is the sintering of metal NPs, occurs due to thermodynamic reasons (i.e., to reduce the surface energy of NPs) and leads to the merging of small NPs into large ones.[18–21] This sintering effect significantly reduces the specific surface area and active sites of metal NPs, ultimately degrading the catalytic performance of supported

[1]School of Physics and Materials Science, Jiangxi Provincial Key Laboratory of Photodetectors, Jiangxi Engineering Laboratory for Advanced Functional Thin Films, Nanchang University, Nanchang, China. [2]School of Materials and New Energy, Ningxia University, Yinchuan, China. [3]State Key Laboratory of Solidification Processing, Center for Nano Energy Materials, Northwestern Polytechnical University and Shaanxi Joint Laboratory of Graphene (NPU), Xi'an, China. [4]Beijing National Laboratory for Molecular Sciences, Organic Solids Laboratory, Institute of Chemistry, Chinese Academy of Sciences, Beijing, China. ✉e-mail: zhangzhouyang@nxu.edu.cn; yiranying@nwpu.edu.cn; huangcs@iccas.ac.cn; feilinfeng@gmail.com

nanocatalysts.[22–24] Therefore, minimizing thermal-induced sintering is an urgent and critical challenge for advancing the applications of supported metal nanocatalysts.

Recently, ultrafast heating has emerged as an intriguing high-temperature synthesis method for producing ultrafine metal NPs or even single atoms on a wide range of supports.[25–32] This approach enables flash heating/cooling of metal precursors on supports within extremely short timescales (several tens to hundreds of milliseconds), facilitating the instantaneous formation of metal nanocatalysts with a high loading rate and good dispersity. For example, Tian et al. report the synthesis of high-density and ultrasmall NPs uniformly dispersed on two-dimensional (2D) porous carbon by the direct carbothermal shock pyrolysis of metal-ligand precursors in just ~100 ms;[33] remarkably, this method not only achieves rapid NP formation but also provides sufficient activation energy for the NPs to overcome the crystallization barrier, instantly forming highly crystallized metal NPs.[34–36] Furthermore, an emerging form of pulsed heating that uses a periodic on-off heating pattern has shown promise in stabilizing metal NPs or single atoms. For instance, Hu et al. demonstrate a high-temperature-shockwave method to synthesize and stabilize metal single atoms; they suggest that the activation energy and ultrafast kinetics provided by this method enable the single-atom dispersion and promote the formation of stable bonds with defects on the substrate.[37] Despite its immense potential for controllable synthesis of sintering-resistant nanocatalysts, the mechanisms governing the formation of small, well-crystallized NPs under such kinetics-dominant, thermodynamically nonequilibrium conditions of pulsed heating remain largely unknown. In situ (scanning) transmission electron microscopy, (S)TEM, with its rapid technological advances in simulating practical working environments, has shown the ability to examine nanoscale dynamics on catalysts and other nanomaterials at an ultra-high spatiotemporal resolution.[38–44] It also provides us a potential opportunity to elucidate the real-time sintering dynamics underlying nonequilibrium processes such as pulsed heating.

Herein, by designing in situ (S)TEM experiments, we show a direct comparative observation on dynamic behaviors in the synthesis of graphene-supported Pt NPs (a suitable catalytic system with a clean interface) under ultrafast pulsed heating (heating/cooling rate of 150 °C s⁻¹ to 1000 °C and pulse width of 50 ms) vs. conventional heating (heating rate of 1 °C s⁻¹). Our results microscopically suggest that pulsed heating promotes the formation of uniform Pt NPs on graphene flakes with great sintering resistance, whereas conventional heating leads to severe sintering of Pt NPs. Coupled with density functional theory (DFT) and molecular dynamics (MD) simulations, we further extract that the significant anti-sintering capacity of Pt NPs under pulsed heating originates from a thermodynamically unstable yet kinetically stable state, which restrains the NPs from long-range diffusion on graphene surface as well as sintering; in contrast, conventional heating allows for free migration and coalescence of Pt NPs on graphene surface. Besides, repeated heating pulses facilitate a gradual improvement in the crystallization degrees of Pt NPs and a progressive optimization of Pt/graphene interfaces, culminating in the formation of stable supported nanocatalysts. Our findings provide a valuable guidance for understanding ultrafast heating process and may stimulate a paradigm shift in the synthesis and applications of supported nanocatalysts.

## Results
### Experimental design
To trace atomic-scale sintering dynamics of Pt NPs on graphene flakes upon pulsed heating, in situ (S)TEM heating experiments are proceeded by utilizing a heating holder equipped with a heating E-chip specimen support, which consists of a spiral-like copper electrode on a silicon nitride (SiN$_x$) membrane with ten holes (covered with amorphous carbon film), as schematically shown in Fig. 1a. The heating

process can be initiated by applying a controlled current through the conductive copper spiral. To guarantee a clean interface across Pt/graphene, graphene flakes are initially dispersed on the heating E-chip and then treated by Ar⁺/O²⁻ bombardment. This treatment preserves the structural integrity of the graphene flakes, as confirmed by the high-angle annular dark-field STEM (HAADF-STEM) images (Supplementary Fig. 1, no visible damage is found throughout the process). Subsequently, an appropriate amount of platinum precursor $(NH_3)_4Pt(NO_3)_2$ is applied on the pretreated graphene flakes, as illustrated in Fig. 1b (see "**Methods**" for the details). The sample is then subjected to a programmed pulsed heating process with precise controls over heating/cooling rate, peak temperature, on-off duration, and repetition cycles. As depicted in Fig. 1c, the pulsed heating protocol involves rapid heating/cooling at a rate of 150 °C s⁻¹ and each pulse of 1000 °C for a short duration of 50 ms over a certain number of cycles (the heating pattern is set based on recent reports and the initial 10-pulse condition is selected to complete the decomposition of precursors and to capture the crystallization process of metal NPs). Particularly, the design of ultrafast cooling rate is essential, as the quenching process is critical for stabilizing the metastable structures formed at high temperatures.[45,46] For comparison, another in situ experiment under conventional heating is parallelly conducted, involving a slow heating rate of 1 °C s⁻¹ to 1000 °C and a subsequent cooling rate of 5 °C s⁻¹ to room temperature (RT) (Fig. 1d; the temperature holding duration is the same as the pulsed heating). It should be of special notice that the holding time at elevated temperatures during conventional heating for the preparation or application of nanocatalysts typically lasts for several hours, which is far longer than that in our case. In both heating modes, Pt precursor is expected to be thermally decomposed into Pt atoms or NPs on the graphene support during initial heating. Prior to the in situ experiment, HAADF-STEM images and corresponding elemental mapping for C and Pt (Supplementary Fig. 2) confirm the uniform distribution of Pt precursor on graphene flakes.

### Sintering dynamics of Pt NPs in two heating modes
We first explore the sintering pattern of Pt NPs on graphene flakes under pulsed heating. Figure 2a–d shows the sequential in situ STEM images of Pt NPs on graphene support against the repeated heating pulses (also see Supplementary Fig. 3). Upon the first pulse, Pt precursor rapidly decomposes, yielding dense Pt NPs on the graphene flake. And after ten pulses, the Pt NPs remain their small size (less than 3 nm, 51% of which are 1-2 nm) (see the corresponding size distributions as insets; additional TEM images of the same area are shown in Supplementary Fig. 4a–c, also confirming the small size of Pt NPs). These observations highlight the strong sintering resistance of Pt NPs under pulsed heating. In a sharp contrast, supported Pt NPs which are subjected to conventional heating display significantly weaker sintering resistance; as shown in the sequential STEM images in Supplementary Fig. 5 (with the selected frames shown in Fig. 2e–h) and TEM images in Supplementary Fig. 4d–f, the number of Pt NPs progressively decreases and the size also increases above 500 °C (also refer to the corresponding size distributions as insets), implying a considerable evaporation and sintering of Pt NPs. Although the lattice-resolved TEM images of Pt NPs formed by both heating modes (Supplementary Fig. 6) reveal a consistent lattice fringe spacing of 0.20 or 0.23 nm (corresponding to the Pt (200) or (111) crystal plane, respectively), the Pt NPs formed via conventional heating exhibit significantly larger particle sizes compared to those formed by pulsed heating, indicating more severe particle agglomeration under slow heating conditions. The above observations are the first microscopic evidence for the anti-sintering characteristics of pulsed heating and are well aligned with the previous synthesis efforts.[33,36,37]

Moreover, to quantitatively analyze the above sintering behaviors, we examine the number and average size of Pt NPs during heating

 

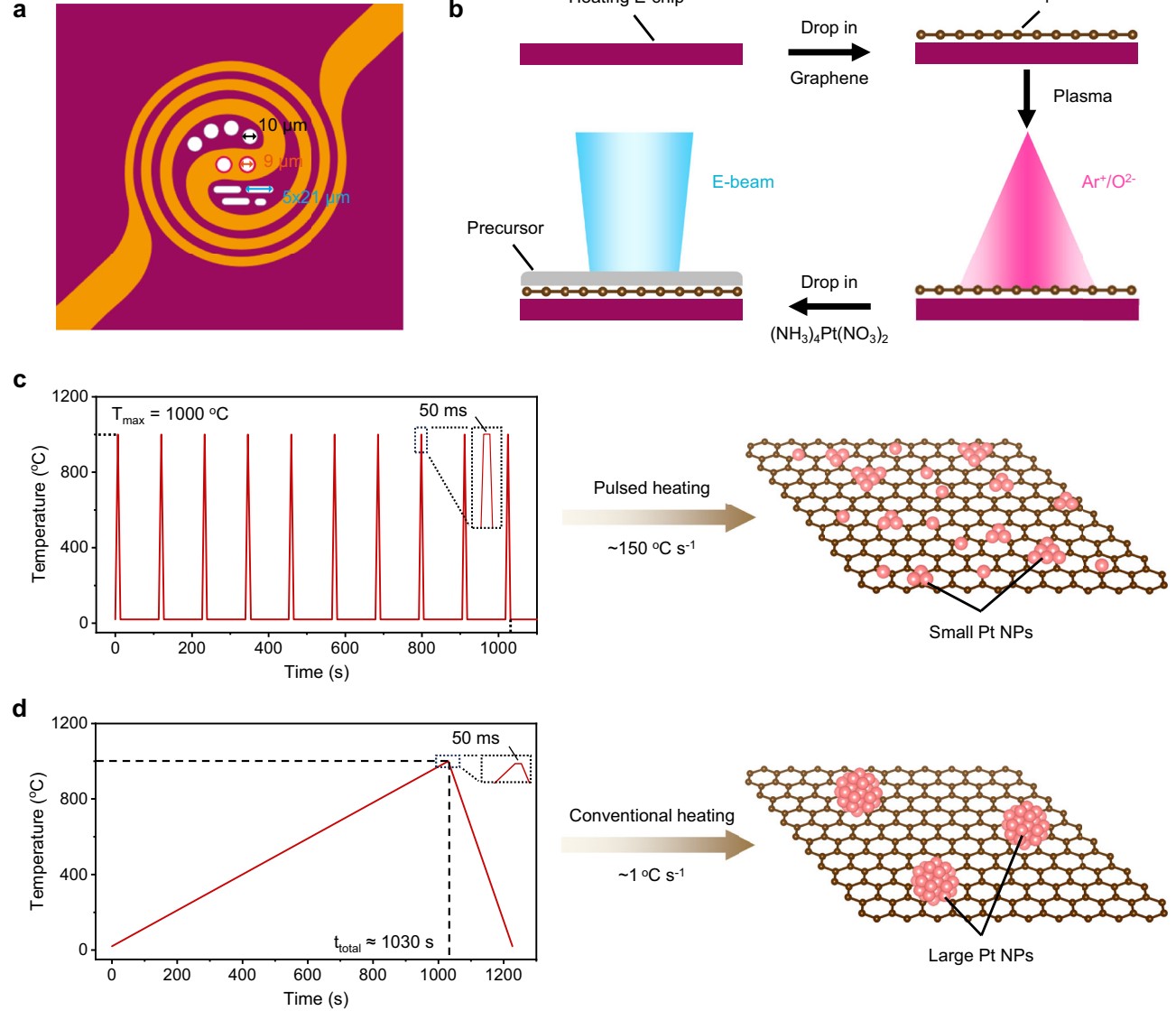

**Fig. 1 | Experimental design for heating Pt NPs in two modes. a** Schematic illustration of heating E-chip. **b** Schematic representation for preparation of (S)TEM specimen, including $Ar^+/O^{2-}$ plasma cleaning of graphene flakes and application of $(NH_3)_4Pt(NO_3)_2$ precursor on them. **c** Temperature profile for pulsed heating and schematic for the product. **d** Temperature profile for conventional heating and schematic for the product.

under the two modes. As shown in Fig. 2i, the number of Pt NPs on graphene under pulsed heating (red line) slowly drops with the increasing pulse number, remaining approximately 71% of the initial number after 10 pulses. In contrast, the number of Pt NPs on graphene under conventional heating (blue line) slowly drops before 500 °C, but the tendency becomes sharp after this temperature; specifically, the number of Pt NPs reduces by about 67% at 500–1000 °C (remaining only 17% after 1000 °C). Besides, Fig. 2j shows the results of particle size vs. heating time/pulses. Pt NPs under pulsed heating (red line) exhibit a slow and steady increase in size, stabilizing at approximately 1.6 nm after 1000 s/10 pulses, which is still within the typical size range (< 3 nm) for high-performance nanocatalysts.[47] Conversely, Pt NPs under conventional heating (blue line) grow rapidly in size until 800 °C, reaching ~1.4 nm. Notably, after 800 °C, a slight decrease in size to around 1.3 nm is observed, which is attributed to the intense diffusion of Pt atoms above the Tammann temperature (namely, diffusion of materials becomes significant above an empirical value of about half of the melting temperature on the Kelvin scale;[48] ~ 750 °C for Pt). At these elevated temperatures, the weak graphene-Pt interaction

allows Pt atoms to diffuse easily, leading to evaporation of small clusters or single atoms, thereby significantly reducing the number of Pt NPs above 800 °C. This explains the particle size-heating time profile observed for the conventional heating. In addition, although the average sizes of Pt NPs are comparable in the above observations, the size distribution from conventional heating is broader due to the existence of many large particles (maximum size up to 5 nm), indicating a more severe sintering. Repeated in situ observations of Pt NPs heated under the two heating protocols further support our interpretation (refer to Supplementary Figs. 7 and 8). We propose that the stark difference in sintering behaviors is primarily attributed to the heating/cooling rate; during the ultrafast pulsed heating, particle migration or atomic diffusion is largely inhibited, whereas the slow heating rate together with the continuous heating in conventional mode allow for more significant atomic diffusion and particle aggregation, which will be discussed later.

To further explore the effect of heating modes on the crystallinity of Pt NPs, which is highly related to the catalytic performance,[49–51] we track the atomic-scale microstructural evolution for individual NPs. As

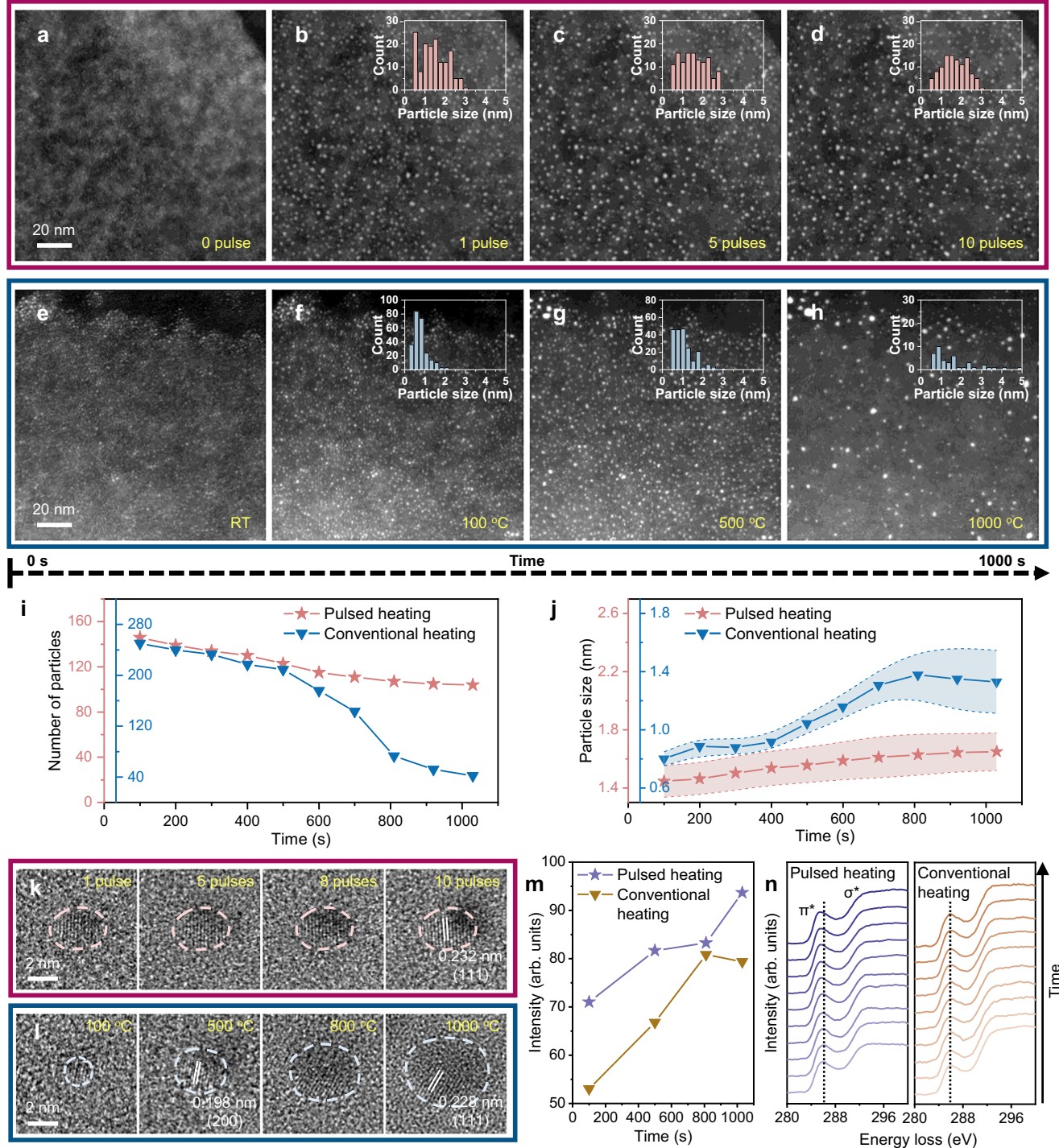

**Fig. 2 | Microscopic sintering behaviors of Pt NPs under two heating modes.**
**a**–**d** In situ STEM image series showing the evolution of Pt NPs on graphene support across 10 heating pulses. Scale bar in (**a**) also applies to (**b**–**d**). **e**–**h** In situ STEM image series showing the evolution of Pt NPs against the rising temperature. Scale bar in (**e**) also applies to (**f**–**h**). The inset in each frame of (**b**–**d**) and (**f**–**h**) showcases the corresponding size distribution of Pt NPs. **i** Number of Pt NPs as a function of heating time under pulsed heating and conventional heating modes. **j** Average size of Pt NPs as a function of heating time under pulsed heating and conventional heating modes. Error bands represent the 95% confidence interval of the mean.

**k**, **l** In situ HRTEM image series showing the structural evolutions of Pt NPs under pulsed heating and conventional heating modes. **m** Evolutions of relative crystalline orders for Pt NPs throughout (**k**) and (**l**) against time. The amounts of crystalline order are determined from the spot intensity of the corresponding FFT patterns (see Supplementary Fig. 9), comparing the value from an NP with the same crystal orientation (whose crystalline order is set as 100 by heating with 10 pulses). **n** C K-edge EELS spectra for graphene support as a function of heating time under pulsed heating and conventional heating modes.

shown in Fig. 2k, in the pulsed heating, Pt NP immediately forms visible (111) lattice fringes after a single pulse and it becomes clearer with the increasing number of pulses, maintaining the exposed crystal plane and the particle size throughout the process (also proved by the

corresponding fast Fourier transform (FFT) patterns in Supplementary Fig. 9a). In contrast, the exposed crystal plane for Pt NP under the conventional heating process changes from (200) to (111) after 500 °C (Fig. 2l) and this NP displays poor crystallinity (see the corresponding

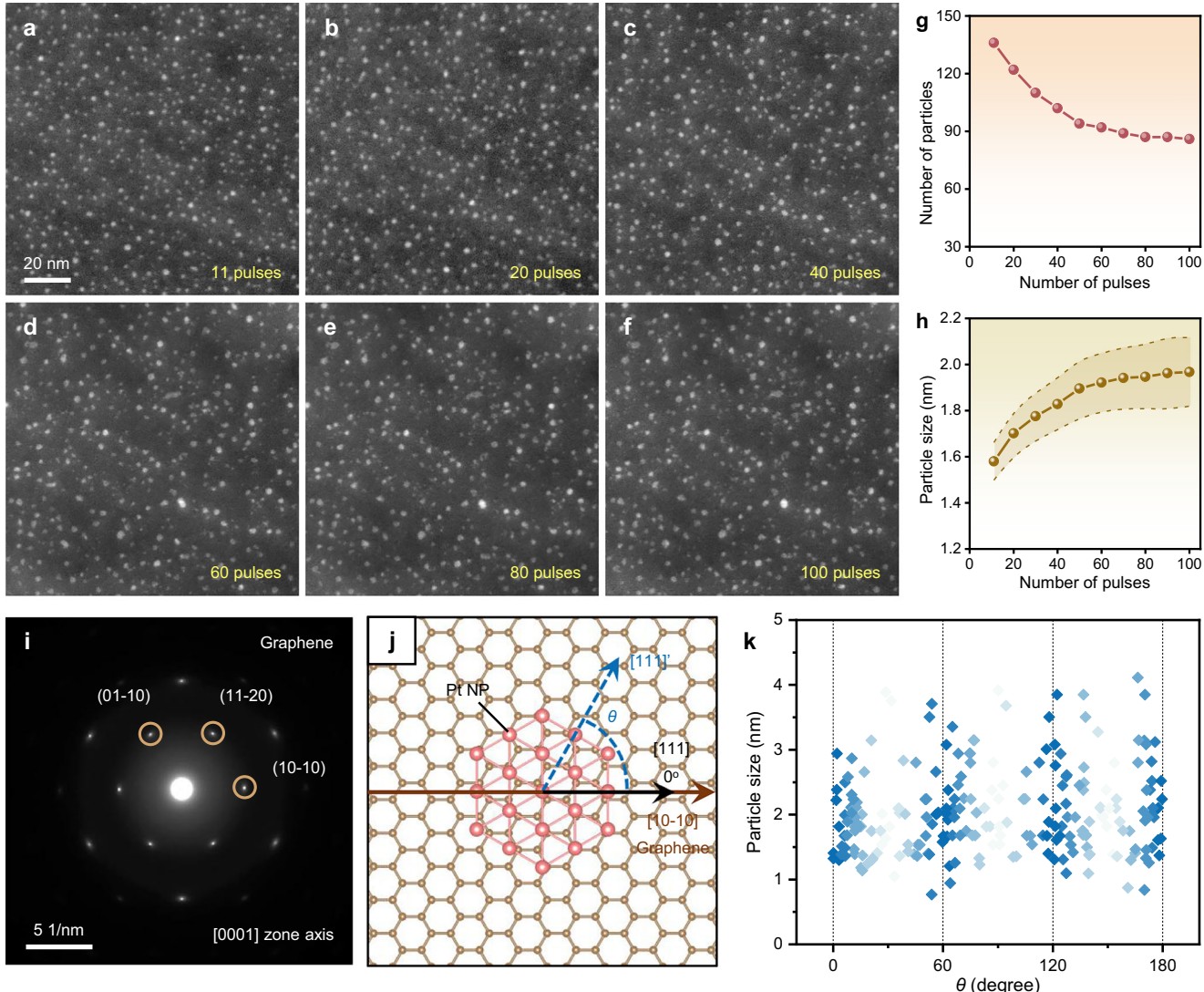

**Fig. 3 | Structural changes of Pt NPs in 100 heating pulses and their crystallographic relationship with graphene support. a–f** STEM image series showing the evolution of Pt NPs on graphene support during 100 heating pulses. Scale bar in (**a**) also applies to (**b–f**). **g** Number of Pt NPs as a function of pulse number across (**a–f**). **h** Average size of Pt NPs as a function of pulse number across (**a–f**). Error band represents the 95% confidence interval of the mean. **i** SAED pattern of the graphene support, showing its single-crystalline nature. **j** Atomic schematic illustrating the orientation of Pt NPs with respect to graphene support. **k** Statistical results of the orientation angles between Pt NPs and graphene support.

FFT pattern in Supplementary Fig. 9b); the underlying reasons for this observation are the continuous size increase (due to the successive merging of nearby NPs) and the thermodynamic-driven shape changes. Figure 2m shows the relative crystalline order of the NPs calculated from their FFT patterns (refer to Supplementary Fig. 9) with respect to time, which further indicate that the NP formed by the pulsed heating possesses a higher crystallinity (the comparison after cooling is also provided in Supplementary Fig. 10 to exclude the temperature difference). Notably, the intensity comparison is also evident at a similar particle size (500 s) for these two heating modes. Besides, to illustrate the evolutions of Pt/graphene interface in the two heating processes, we record the core-loss C K-edge electron energy loss spectroscopy (EELS) spectra for graphene support as a function of time. Each C K-edge spectrum consists of a peak at ~284 eV (the transitions from 1 s spin level to empty π* orbitals of the sp²-bonded atoms) and a step at ~289 eV (the excitations to empty σ* orbitals at both sp² and sp³ bonded atoms).[52] As shown in Fig. 2n, the π* excitation of C K-edge gradually shifts toward lower energy loss values during pulsed heating (noticeable shift begins at the 4th pulse, likely due to

the close contact between NPs and support), which can be a result of strong bonding and charge transfer between Pt NPs and graphene support that modify the electronic structure around the active sites; yet it hardly shifts in the conventional heating. These results show that strong metal-support interactions are established in Pt/graphene interface during pulsed heating, which is of key importance to the stability for supported nanocatalysts.[53,54]

## Atomic-scale behaviors of Pt NPs in pulsed heating

As the sintering rate of Pt NPs remains nearly linear during the first ten pulses of heating, we further extend the pulse number to 100 for understanding the effect of pulses on the sintering stability of Pt NPs. Sequential in situ STEM images (Fig. 3a–f, also see Supplementary Fig. 11) show that, even after 100 pulses, Pt NPs undergo slight sintering, maintaining a nanoscale size and nearly the same particle distribution throughout the heating process. The elemental mappings and the line scan together with the corresponding HAADF-STEM images of Pt/graphene interface are provided as Supplementary Fig. 12, indicating the dense and synchronous distribution of Pt NPs on

graphene flake after 100 pulses. Furthermore, the number and average size of Pt NPs during 100 pulses are also analyzed as functions of pulses, as shown in Fig. 3g, h. Both curves behave a tendency to flatten out as the number of pulses increases, ultimately stabilizing with an average size of 2.0 nm and a particle number retention of 63%; this indicates that the Pt NPs are gradually stabilized on graphene support after repeated heating pulses. In addition, the C K-edge EELS results of graphene support before and after 100 heating pulses confirm its structural integrity under repeated heating pulses (Supplementary Fig. 13).

It is also worth noting that the morphology of Pt NPs evolves during the pulsed heating process (Fig. 3a–f), showing a gradual development from quasi-spherical to faceted NPs (high-resolution TEM (HRTEM) images of the Pt NPs after 100 pulses in Supplementary Fig. 14 exhibit hexagonal NPs with dominant {111} facets and occasional (200) planes). This observation reflects that the Pt NPs undergo crystallization improvement and structural optimization on graphene support during the pulsed heating, which are crucial for the stability of nanocatalysts. To clarify the above process, we further analyze the orientation relationship between Pt NPs and graphene flake after 100 pulses, as metal particles tend to form an epitaxial orientation on 2D supporting materials during their growth.[55,56] Selected-area electron diffraction (SAED) pattern of a graphene support is shown in Fig. 3i, displaying its single-crystalline nature along with [0001] zone axis, and the corresponding atomic structure is schematically shown in Fig. 3j. To define the relative orientation of Pt NPs against graphene flakes, we set the angle between them as 0° when the graphene [10-10] direction (brown line) aligns with the Pt [111] direction (black line). When the Pt NP orientation (blue dashed line) deviates from the graphene orientation, the angle is denoted as $\theta$, as shown in Fig. 3j and Supplementary Fig. 15. Then, we measure the $\theta$ angles of approximately 200 Pt NPs, as statistically shown in Fig. 3k. We find that the distribution of $\theta$ angles is mainly around 0°, 60°, 120°, and 180°, indicating that the Pt NPs align their crystallographic orientations with graphene support during pulsed heating. This high degree of orientational alignment between Pt NPs and graphene support minimizes the interfacial energy, thereby driving the formation of a thermodynamically stable structure.

In order to directly disclose the structural optimization of Pt NPs, we accordingly capture the HRTEM images of graphene-supported Pt NPs with representative structures along with the increasing heating pulses. As shown in Fig. 4, the structural optimization of Pt NPs can be divided into three stages. In the first stage (during the first 10 pulses, Fig. 4a, b), the Pt NPs, which are newly formed from the decomposition of the precursor, are irregularly elliptical in shape with a poor crystalline nature (displaying only the (200) plane). In the second stage (-10–40 pulses, Fig. 4c, d), Pt NPs slightly grow and form an elongated hexagonal shape with well-defined crystalline lattices, resulting from the actions of facet development. In the final stage (up to 100 pulses, Fig. 4e–h), for minimizing the system energy, Pt NPs undergo persistent surface atomic rearrangement and ultimately form a regular hexagonal outline. The corresponding inverse FFT (iFFT) patterns show their distinct atomic arrangements (Fig. 4i-l), and the entire structural evolution of Pt NPs on graphene is schematically shown in Fig. 4m. In contrast, Pt NPs under conventional heating generally retain an irregularly elliptical shape (Supplementary Fig. 16), which is likely due to the slower heating kinetics; in this scenario, NPs have sufficient time for atomic diffusion and particle agglomeration, leading to the formation of thermodynamically stable spherical structures. Conversely, the rapid heating kinetics under pulsed heating suppress such agglomeration; and aided by the interfacial effects of graphene substrate, the pulsed heating finally results in the formation of faceted structures. Again, repeated in situ observations of Pt NPs sustained 100 heating pulses further support this deduction (refer to Supplementary Fig. 17). It is worth mentioning that the above structural evolution process emphasizes the advantages of pulsed heating in modulating

the exposed crystal plane of metal nanocatalysts. It also demonstrates a distinct, particle-centered, and kinetic-controlled pathway in which pulsed heating drives stepwise structural optimization of NPs, in contrast to previous reports where a single high-energy pulse induces support reconstruction and defect generation to stabilize NPs.[35]

In addition, to assess the thermal stability of the Pt/graphene nanocatalyst formed after 100 heating pulses, we conduct an extended heating test by maintaining it at 1000 °C for 1 h. As shown in Supplementary Fig. 18, no obvious coarsening or change in particle morphology occurs within the observation, demonstrating that the as-produced nanostructures are robust even under continuous high-temperature exposure. The marked sintering resistance can be attributed to the formation of strong Pt-C interfacial bonding and the progressive structural alignment of NPs with graphene substrate, both of which arise during the prior heating pulses. To demonstrate that pulsed heating stabilizes NPs through a kinetically constrained pathway rather than a short high-temperature exposure, we further perform an extended pulsed heating of 1000 pulses with a 1 s peak duration (cumulative exposure of 1000 s at 1000 °C). Again, Pt NPs remain stable without observable sintering (Supplementary Fig. 19), and the evolutions for number and average size of Pt NPs during the first 100 pulses follow similar trends as those in Fig. 3.

## Sintering kinetics of pulsed heating

Summarizing from the above results, we infer that the enhanced sintering stability of Pt NPs under pulsed heating is attributed to two main factors: first, the as-revealed structural optimization and orientational alignment of Pt NPs during pulsed heating lead to an energetically stable configuration; and second, NPs are kinetically hindered from undergoing effective migration due to the brief duration of heating pulses, which is the key mechanism underlying their sintering resistance. To further support this hypothesis, DFT calculations are thereby performed. Figure 5a–c display the DFT calculated patterns of Pt NPs with (001), (011) or (111) planes bound onto graphene. The corresponding binding energies are also shown in Fig. 5d, from which it can be concluded that the Pt(111) NP exhibits the lowest binding energy among the three configurations; this indicates that the attachment of Pt NPs on graphene flakes is more energetically favorable at their (111) facets. This is the direct reason that Pt/graphene interfaces are gradually optimized and stabilized during the pulsed heating (Fig. 4).

Moreover, in order to comprehend the kinetic aspects for sintering behaviors of Pt NPs under the two heating modes, we further perform MD simulations on a Pt/graphene system with different temperature profiles. Considering the difference in time scales between simulations (usually on a picosecond scale) and experiments, we use two distinct heating processes: a temperature ramping process (from 25 to 1000 °C over 20 ps, Supplementary Fig. 20a) to model the pulsed heating experiment and a temperature holding process (at 1000 °C for 20 ps, Supplementary Fig. 20b) to simulate the conventional heating process due to its extremely slow heating rate compared to the pulsed heating. As a result, the system energies as functions of the simulation time for these two processes show their remarkable differences (Supplementary Fig. 20), in which the temperature ramping process shows a gradual increase in energy while the temperature holding process starts at a higher energy state, and the average energy of the former is obviously lower than that of the latter. The simulations also clearly depict the structural evolutions of Pt NPs under two heating methods (Fig. 5e, f); in the temperature ramping process, the Pt NP maintains its integrity on graphene support without apparent diffusion or migration, while a significant diffusion of Pt atoms occurs and the NP eventually wets the graphene surface in the temperature holding process. Although this result clearly suggests that ultrafast heating leads to a mitigation for sintering, it also implies that the NPs can still be thermodynamically unstable as their structure and position both change (Fig. 5e). This means that the as-observed sintering

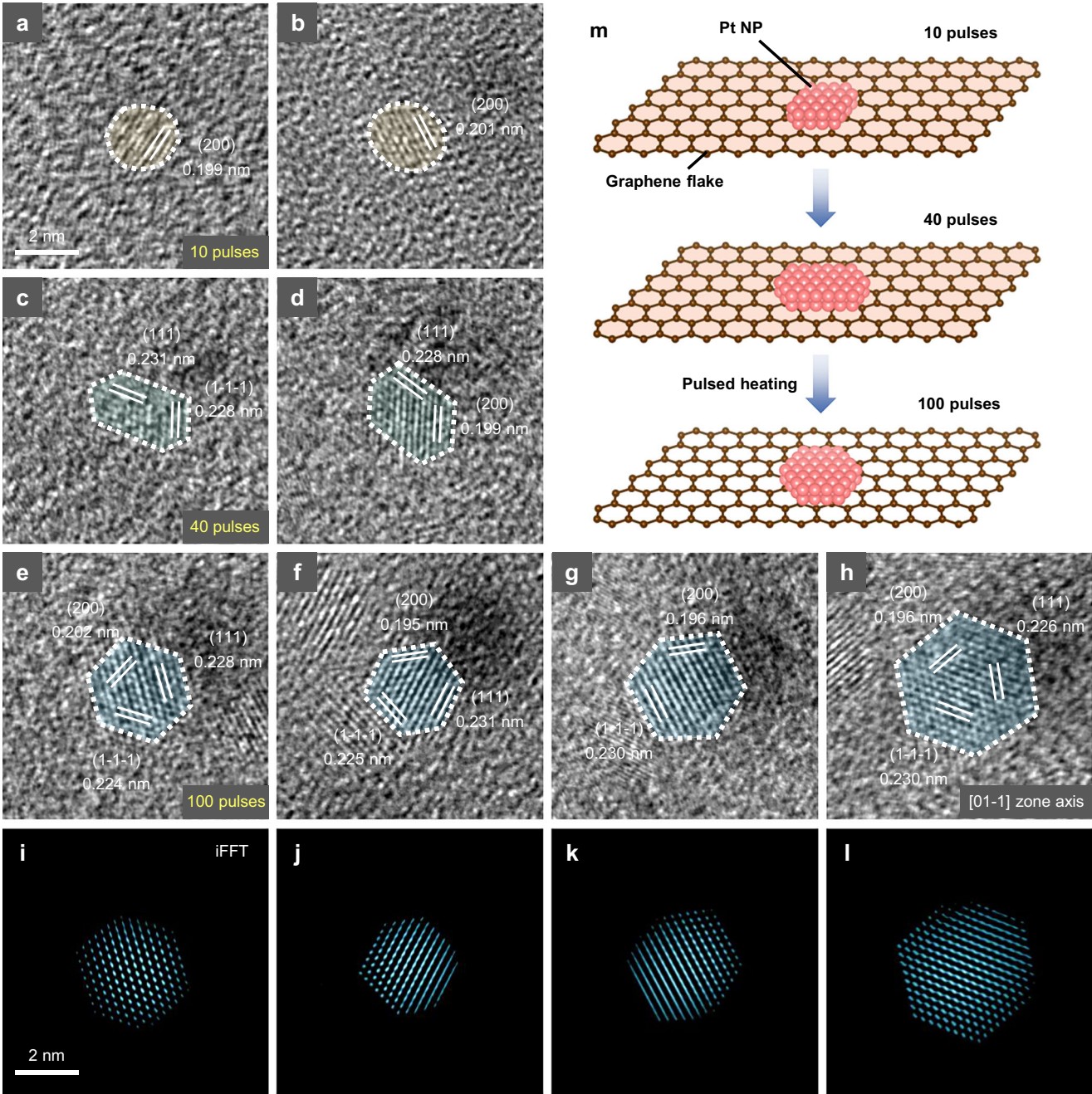

**Fig. 4 | Atomic-scale evolutions of Pt NPs in 100 heating pulses.** HRTEM images showing the lattice structures of representative Pt NPs after (**a**, **b**) 10 pulses, (**c**, **d**) 40 pulses, and (**e–h**) 100 pulses. Scale bar in (**a**) also applies to (**b–h**). (**i–l**) False-colored iFFT patterns showing the atomic arrangements of Pt NPs corresponding to (**e–h**), respectively. Scale bar in (**i**) also applies to (**j–l**). **m** Schematic illustration for structural evolution of Pt NPs on graphene support via pulsed heating.

resistance for pulsed heating is not solely induced by thermodynamic causes, kinetic factors may also play important roles.[34,37] To further verify the kinetic stability of Pt NPs, we track their sintering behavior during the first pulse by rapidly heating them to 1000 °C and then holding for a duration (Fig. 5g). At this early stage, the interaction between the NPs and the support has not yet developed. It is then found that no noticeable sintering occurs at the beginning of temperature holding (< 6 s; be noted that this time is much longer than the pulse width); nonetheless, after that, sintering dramatically occurs (Fig. 5h). The finding well agrees with the recent report by Zeng et al. (the critical relaxation threshold for sintering is ~10 s),[57] which appropriately indicates that Pt NPs are kinetically stable in the case of short heating pulses.

## Discussion

In general, the fundamental reason for sintering of supported NPs arises from the competitive relationship between the surface energy of NPs and the interfacial energy of system; small NPs have low interfacial energy yet high surface energy (tending for sintering) while large NPs are dominant by their high interfacial energy with the support (tending for stabilization). Therefore, upon conventional heating with a slow heating rate and prolonged holding time, supported NPs are free to migrate and coalesce with other NPs, which is the major issue of nanocatalysts as sintering. In contrast, under pulsed heating, although small NPs become thermodynamically unstable at high temperature, the ultrafast heating/cooling rate together with the short pulse width make the sintering process kinetically impossible. In other words,

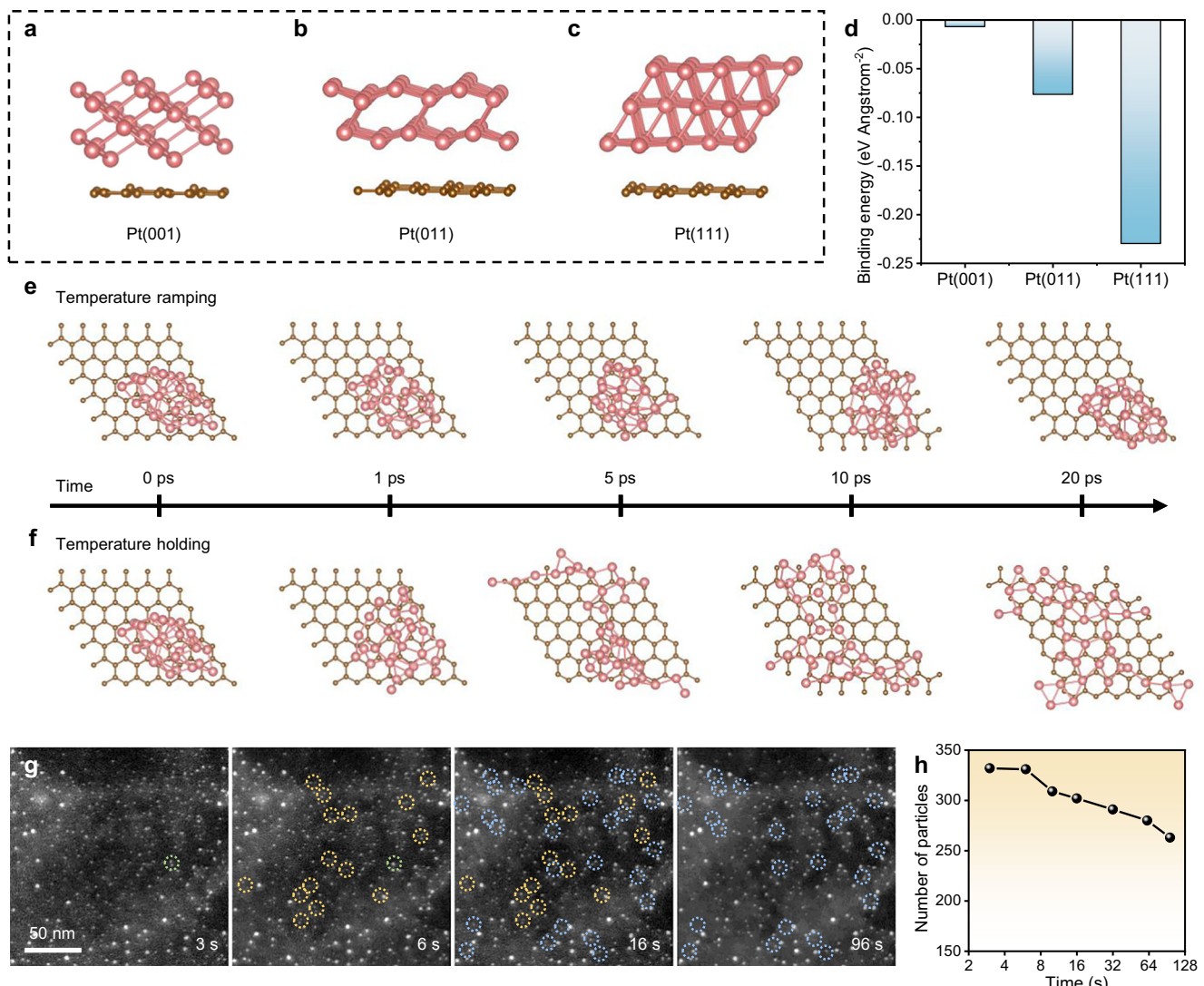

**Fig. 5 | DFT calculations and MD simulations for sintering dynamics.**
**a–c** Binding patterns and (**d**) corresponding binding energies of Pt(001), Pt(011), and Pt(111) clusters on a graphene flake. **e, f** Snapshots during MD simulations (0, 1, 5, 10, and 20 ps) for a Pt/graphene structure with different temperature profiles. Color code for atoms: C, brown; Pt, pink. Maximum bond length criteria: Pt-Pt, 0.

281 nm; Pt-C, 0.237 nm. **g** Time-lapse STEM images of graphene-supported Pt NPs after rapid heating (150 °C s⁻¹) to 1000 °C. Colored cycles highlight the sintering events of NPs after 6 s (green), 16 s (yellow), and 96 s (blue). **h** Number of particles as a function of time for (**g**).

supported metal NPs are actually in a metastable state under pulsed heating, which is thermodynamically unstable but kinetically stable; therefore, large-scale atomic diffusion and particle migration, which are the primary driving forces of sintering, are effectively inhibited. Consequently, our work offers clear mechanistic insights that NP stabilization for a specific metal during pulsed heating can be achieved within a kinetic window (which should be jointly governed by heating/cooling rate (Supplementary Figs. 21 and 22), pulse duration (Supplementary Fig. 23), and peak temperature, as discussed in **Supplementary Discussion**); under such conditions, diffusion-driven sintering is kinetically confined while local structural rearrangements are activated, enabling the formation of thermally robust nanostructures.

On the basis of the above analysis, we finally discuss the microstructural advantages of those pulsed heating-derived NPs for catalytic applications: 1) the ultra-small and highly dispersed NPs provide abundant active sites, which are promising for improving the mass-specific catalytic activity;[22,47] 2) the high-crystalline NPs together with the metastable metal/support structure ensure high intrinsic activity as well as excellent sintering resistance;[49,58,59] and 3) the consequential

metal-support interaction (i.e., charge transfer) enhances substantial interfacial stability and modulates the electronic structure of metal NPs, which are beneficial for long-term catalytic stability and reaction selectivity.[53,54,60] The synergistic effect of these structural advantages directly underpins the catalytic performance and long-term stability of these catalysts, demonstrating the great potential of pulsed heating for precise and facile regulation of supported nanocatalysts.

In summary, our systematic experimental and theoretical study microscopically reveals that pulsed heating significantly mitigates the thermal-induced sintering of graphene-supported Pt NPs, in contrast to the severe sintering observed from conventional heating. We find that the sintering resistance of Pt NPs arises from their thermodynamically unstable yet kinetically stable state under pulsed heating, which suppresses long-range diffusion and sintering of NPs on graphene surface; in contrast, conventional heating allows for free migration and coalescence of Pt NPs on graphene surface. As a result, repeated heating pulses drive progressive improvement in the crystallization degrees of Pt NPs and promote their interfacial optimization with graphene support, leading to structurally refined and thermally

robust supported nanocatalysts. These results provide mechanistic understanding of pulsed heating in stabilizing metal NPs and may offer valuable insights into the nonequilibrium synthesis of high-performance nanocatalysts.

## Methods

### Precursor preparation

A 5 mL graphene oxide solution was initially freeze-dried to form solid blocks, which were subsequently reduced under $H_2$ atmosphere at 500 °C for 5 h to obtain graphene blocks. The graphene blocks were dispersed in ethanol solution and sonicated for 20 min to produce the precursor of graphene flakes. The precursor was dropped onto a heating E-chip specimen support and dried under ambient conditions. The graphene surface was then treated using a Gatan Solarus II plasma cleaner, subjected to $Ar^+/O^{2-}$ bombardment at a flow rate of 30 sccm for 60 s. The system pressure was maintained at 350 mTorr, with the plasma power set to 40 W. Next, ammonium platinum nitrate $(NH_3)_4Pt(NO_3)_2$ was dissolved in ethanol to prepare a Pt precursor solution, which was sonicated for 20 min prior to use. Finally, the Pt/graphene precursor was prepared by drop-casting the Pt precursor solution onto the graphene-coated heating E-chip.

### In situ (S)TEM experiments

In situ (S)TEM experiments were performed using a DENSsolutions Wildfire heating holder. TEM experiments were conducted on a JEOL JEM-F200 cold-field-emission TEM operated at 200 kV, equipped with an EDS and a Gatan Continuum S EELS. TEM images were acquired using a Gatan OneView IS camera with an optimized exposure time. STEM images were captured with a Gatan PEELS camera using an exposure time of approximately 30 s (except for the images in Fig. 5g, which had an exposure time of 2 s). Core-loss EELS spectra were collected in STEM mode; the energy resolution was 0.72 eV with a dispersion of 0.15 eV per pixel and the acquisition time was about 1 s per frame (total acquisition time ≈ 20 s). With this setup, the convergence and collection angles were 0 and 100 mrad, respectively. For pulsed heating experiments, the sample was subjected to a heating/cooling rate of 150 °C s$^{-1}$ to 1000 °C for 50 ms, and (S)TEM images were acquired at RT after each pulse (separation of pulses: 100 s). In conventional heating experiments, the sample was heated from RT to 1000 °C at a ramping rate of 1 °C s$^{-1}$; and the cooling rate was 5 °C s$^{-1}$. All in situ experiments were repeated to ensure reproducibility of the structural evolution dynamics. The electron beam was blanked whenever possible to minimize beam effects on the nanocatalysts.

### Computational details

All theoretical calculations were performed with the Vienna Ab initio Simulation Package (VASP, version 5.4.4) code.[61] DFT calculations were performed with projector-augmented wave method,[62] and the Perdew-Burke-Ernzerhof method of generalized gradient approximation was used to describe the exchange-correlation functional.[63] Grimme's DFT-D3 method was applied for the van der Waals corrections.[64] Energy and force convergence criteria were set as $10^{-5}$ eV and 0.02 eV Å$^{-1}$, respectively. A 6×6 matrix was used to model graphene and a vacuum layer with a thickness of more than 15 Å was added to avoid spurious interactions between cells. Ab initio MD simulations were performed in the NVT ensemble with Nosé-Hoover thermostat,[65,66] and the time step was set as 2 ps.

## Data availability

All data that support the findings of this study are presented in the article and Supplementary Information. Data are available from the corresponding authors upon request. Source data are provided with this paper.

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

## Acknowledgements

L.F. acknowledges the support from the National Natural Science Foundation of China (22375081 and U21A20500) and the Nanchang

University. C.H. acknowledges the support from the National Key Research and Development Project of China (2022YFA1204500 and 2022YFA1204501) and the Strategic Priority Research Program of the Chinese Academy of Sciences (XDB0520200). Z.Z. acknowledges the support from the National Natural Science Foundation of China (22405142) and the Natural Science Foundation of Ningxia (2024AAC03027). Y.Y. acknowledges the support from the National Natural Science Foundation of China (22502157). J.H. acknowledges the support from the Jiangxi's Creative Special Fund for Graduate Students (YC2024-B037).

## Author contributions

L.F. and C.H. conceived idea for the project. L.F. and J.H. designed experiments. Y.Z., J.Z. and C.X. synthesized TEM samples. J.H., Z.Z., J.C. and G.W. conducted TEM observations and data analysis. Y.Y. performed DFT calculations and MD simulations. J.H., Z.Z., Y.Y., C.H. and L.F. wrote the manuscript with the help and input from all authors. All authors have given approval to the final version of the manuscript.

## Competing interests

The authors declare no competing interests.
