## [Transparent Peer Review file · Nature Communications]

Nonequilibrium pulsed heating freezes sintering of supported metal nanocatalysts

Corresponding Author: Professor Changshui Huang

Version 0:

Reviewer comments:

Reviewer #1

(Remarks to the Author)

In the manuscript, Huang et al. report ultrafast pulsed heating as a nonequilibrium strategy to suppress the sintering of supported Pt nanoparticles, using graphene as the support. Through a combination of in situ STEM analysis, EELS, DFT, and MD simulations, the authors reveal that pulsed heating induces a thermodynamically unstable yet kinetically stable state, enabling improved crystallinity and enhanced metal–support interactions while effectively inhibiting particle migration and coalescence. This work provides a novel atomic-scale understanding of metastable states in nanocatalysts and offers an alternative route to sintering-resistant catalyst synthesis beyond thermodynamic constraints. Overall, the authors present a solid and well-validated study, and this manuscript will be of high interest to researchers working on nanocatalysts. The methodology is rigorous, and the mechanistic insights are valuable. This manuscript can be considered for publication in Nature Communications. However, the following points should be addressed before publication to further improve the quality of this work.

1. The authors mention that the heating pattern was designed based on recent reports; however, a more detailed description of the temperature profile, particularly the cooling rate following each pulsed heating, is necessary. Since cooling kinetics are critical for the formation of metastable nanostructures, quantifying the cooling behavior would greatly enhance the mechanistic understanding. Additionally, the selection of the 10-pulse condition as the primary reference over the 1-pulse case remains unclear. Could the authors clarify and comment on the potential effects of a single, higher-temperature pulse?

2. The authors mention that ultrafast pulsed heating effectively suppresses atomic diffusion and aggregation of nanoparticles. However, in several recent studies, including direct-joule heating-based redispersion/atomization (Journal of the American Chemical Society 142,41, 2020, 17364-17371., Science 359, 6383, 2018, 1489-1494.), and photothermal effect-driven redispersion/atomization (metal oxide redispersion (Advanced Materials, 2025, 2419790.), Pt atomization (Chem 8, 4, 2022, 1014-1033.)), it has been shown that ultrafast high-temperature treatment can actually promote atomic diffusion while simultaneously restricting aggregation, resulting in smaller or even atomically dispersed particles. Could the pulsed heating strategy used in this study also induce redispersion or partial atomization of Pt nanoparticles, especially at elevated pulse temperatures or with increased pulse counts? In such cases, atomic mobility could be activated while aggregation remains kinetically suppressed due to short exposure durations. The authors are encouraged to comment on whether such size reduction or dispersion behavior could emerge under more intensive pulse conditions and whether this pathway has been experimentally tested or theoretically predicted.

3. The manuscript reports the application of up to 100 successive heating pulses, yet no notable structural or chemical change in the graphene support is discussed. Could the authors elaborate on the methods used to evaluate the integrity of the graphene after repeated thermal treatments? In particular, were any atomic-level characterizations (e.g., Raman spectroscopy, EELS, or aberration-corrected TEM) conducted to assess possible cumulative damage or defect generation?

4. The discussion compares the appearance of the Pt (111) facet after a single pulsed heating step with the delayed emergence of the same facet under conventional heating (initially showing the (200) facet). Given that (111) is thermodynamically more stable than (200), it would be more accurate to highlight the rapid stabilization of (111) as a merit of pulsed sintering rather than concluding a higher degree of crystallinity. Moreover, as Figure S12 also shows the appearance of the (200) facet under pulsed heating, please address this apparent contradiction.

5. A noticeable shift in the peak position begins at the 4th pulse. Please provide a discussion on this phenomenon. Additionally, prior to comparing 10-pulse treatment with conventional heating for 1000 seconds, consider providing comparative data at intermediate temperatures to better contrast single-pulse effects.

6. In the interpretation of Figure 3j, clarify the rationale for lattice matching discussion between 3D Pt nanoparticles and 2D graphene sheets. In particular, provides more detail on how the crystallographic orientation was determined from FFT patterns.

7. Only four images are shown for the 100-pulse case in Figure 4(e-h). Please revise the layout for consistency across all pulse conditions or clarify the rationale behind the image selection.

8. Confirm whether the TEM image in Figure S13 was obtained along the same zone axis as those shown in Figure 4. This is important for accurate structural comparison.

Reviewer #2

(Remarks to the Author)

The manuscript titled “Defying thermodynamics: Nonequilibrium pulsed heating freezes sintering of supported metal nanocatalysts” explores the behavior of Pt nanoparticles supported on graphene under ultrafast pulsed heating, using in situ (S)TEM observations in combination with DFT and MD simulations. The study aims to demonstrate that pulsed heating leads to a metastable, kinetically stabilized state that can inhibit nanoparticle sintering more effectively than conventional heating. While the experimental design is technically sound and the integration of microscopy with modeling is methodologically appropriate, the current version of the manuscript raises several concerns that must be addressed before it can be considered for publication. The mechanistic claims, particularly regarding structural optimization, interface effects, and the generality of the proposed anti-sintering mechanism, require clearer justification and more critical discussion. Additionally, the manuscript would benefit from a more rigorous evaluation of the crystallinity data, the role of the substrate, and the absence of catalytic performance validation. Therefore, I recommend major revisions. Specific comments are provided below to help the authors improve the manuscript’s clarity, depth, and scientific rigor..

1. In Figure 2m, The lower crystallinity of Pt NPs under conventional heating may be biased by the different thermal states during which the measurements were taken—if FFT was performed at high temperature, thermal vibrations and atomic diffusion could artificially reduce apparent crystallinity. In contrast, pulsed heating benefits from repeated cooling, effectively annealing the structure. A fair comparison should assess crystallinity after cooling in both cases, or this limitation should be clearly discussed to avoid misinterpretation. Including such a control or at least discussing this thermal-state-dependent artifact would significantly enhance the credibility of the crystallinity comparison and deepen our understanding of the thermal evolution of NP structure in both regimes.

2. In Figure 2n, the electron energy loss spectroscopy (EELS) results suggest changes at the metal-support interface, but the chemical state evolution of Pt (e.g., oxidation state, bonding environment) is not fully explored. Complementary characterization would help clarify the electronic effects contributing to NP stabilization.

3. In Figure 3, the manuscript claims a “high degree of lattice matching” between Pt NPs and graphene supports. However, this terminology is not strictly accurate. Pt(111) has a lattice spacing of ~0.227 nm, while the in-plane lattice spacing of graphene (e.g., (10-10)) is ~0.213 nm, and their crystallographic symmetries (fcc vs. hexagonal) differ. Thus, this interface should not be described as lattice-matched. A more appropriate term would be “orientation alignment” or “epitaxial-like interfacial relationship.” The authors are advised to revise this statement for precision.

4. In the part of “Sintering kinetics of pulsed heating”, The manuscript attributes the sintering resistance under pulsed heating to both interfacial structural optimization and kinetic suppression. It would strengthen the mechanistic understanding if the authors discuss whether kinetic limitation alone—without crystallographic alignment or structural optimization—would be sufficient to ensure long-term stability. This distinction is important to evaluate whether the observed robustness is primarily due to the heating profile or also critically dependent on support-induced ordering

5. Although 100 pulses demonstrate improved sintering resistance, the net thermal exposure remains short relative to real catalytic conditions. The authors are encouraged to address how these metastable structures behave under extended heating, cyclic heating, or in the presence of reactive environments, either through additional data, cited references, or detailed discussion.

6. While structural stabilization is clearly demonstrated, the manuscript lacks direct correlation with catalytic performance. A brief experimental validation (e.g., a model reaction demonstrating enhanced durability or activity of pulsed-heated Pt NPs) or a robust discussion on how the observed microstructural advantages may translate into catalytic improvements would significantly enhance the relevance of the study.

Version 1:

Reviewer comments:

Reviewer #1

(Remarks to the Author)

This study played a crucial role in elucidating the mechanism for rational nanostructured catalyst design through heat treatment in a metastable state.

Furthermore, I believe it is fully acceptable for accurately answering the questions posed.

Reviewer #2

(Remarks to the Author)

Many concerns have been addressed. However, after careful review of the revised manuscript and rebuttal, there are still major questions for this paper.

1. What dose the authors mean about “defying thermodynamics”. How did you achieve that and how can anyone defy thermodynamics? A mere anti-sintering behavior is not defying thermodynamics.
2. The particle formation and migration process have been previously studied by Nat. Commun. 2020, 11 (1), 6373, and others.
3. What is the safe region for ultrafast heating to maintain particle stability, for example, T range, t or pulse times range, and heating/cooling ranges. The authors should dig into the mechanism of this kinetic stabilization and propose design principles, instead of just listing one example.
4. The authors used stability under pulsed heating as improved stability and sintering-resistance. This is far from enough as thermochemical reactions occur at continuous high T rather than pulses. The true isothermal stability is therefore important to evaluate if the particles are stable or not.
5. The conclusion—that pulsed heating achieves a higher degree of crystallinity than conventional heating—should be discussed more cautiously. A conventional heating and a slow cooling provide sufficient time for atoms to diffuse and rearrange into their lowest-energy, most stable lattice positions, thereby minimizing defects and yielding a highly ordered structure. This process (i.e., annealing) is the widely accepted standard method for improving the crystallinity of materials.

Version 2:

Reviewer comments:

Reviewer #2

(Remarks to the Author)

The authors did not really address the previous concerns. As shockwave heating to stabilize nanoparticles or nano clusters has been well documented before, using the exact shock method. Now the authors re-performed the experiments under TEM, which, to some degree, provides a fresh perspective. However, if this nanoscale perspective is not well-utilized for an in-depth understanding, then little science would be gained from the current study. Importantly, the origin of such kinetic stabilization is still not revealed, and many of the experimental designs are not scientifically sound or even conflict with each other. To name a few:

1. There is a significant discrimination between major control samples (Fig.1c vs. Fig.1d). They have very different thermal histories at high T, and, understandably, they show different dispersions. Instead of controlling the total time (~1000 s or only 10 pulses), the authors should do 10,000-20,000 pulses (500-1000s at high temperature) for a fair comparison. The so-called kinetic stability may not be valid because of the kinetic advantages, but may largely be from a much much shorter duration at high temperature. This can sabotage the foundation of this paper.

2. In the last round, the authors were asked to test the isothermal stability of Pt/graphene. They responded that this material can sustain 1000°C for 1 hour without sintering (reply 4, Fig. S18). This result conflicts directly with Fig. 5g-h where nanoparticles show sintering starting at 6 seconds.

The reviewer tends to believe that the latter is the true case, as sintering is much easier at 1000°C, where nanoparticles want to sinter and the graphene substrate is graphitized. As long as you do not have strong Pt-C/O bonds, the sintering is almost unavoidable. In the paper, the authors discussed having some crystalline affinity of Pt and the graphene, but no direct bond formation. Then the Pt clusters can not be stabilized at such a high temperature. This will be contrary to common sense.

Fig. 5g-5h showing sintering in 6 seconds at 1000°C:

Reply and Fig. S18 showing anti-sintering in 1 hour at 1000°C:

3. This question is not answered at all in the last round.--- What is the safe region for ultrafast heating to maintain particle stability, for example, T range, t or pulse times range, and heating/cooling ranges? The authors should dig into the mechanism of this kinetic stabilization and propose design principles, instead of just listing one example. We understand the current pulse enables stability under 10 pulses. What we don't know is where the kinetic boundary is and the reason behind this boundary, for example, the kinetic competition between interparticle diffusion and the pulsed heating freezing. The author also mentioned heating rate is particularly important, but is this supported by experimental proof (must with fair controls)? Quantitative or numeric study will be largely preferred to understand this kinetic stability or its safe boundaries.

Version 3:

Reviewer comments:

Reviewer #2

(Remarks to the Author)

The authors have made a substantial effort to address the concerns raised in the previous round by providing more in situ TEM evidence. Overall, the revised manuscript is largely improved in rigor and scope, and it convincingly demonstrates that pulsed heating can yield supported Pt nanoparticles with markedly enhanced resistance to sintering. However, one key concept and (mis)understanding should be clarified.

1. It remains insufficiently clarified on the roles of pulsed heating (PH) and Pt–C interfacial bonding in governing the long-term stability of the nanoparticles. In comment #2, the authors showed that, under PH operation, Pt NPs are not stable for the initial stage because their crystallinity and interfacial bond are not yet formed; The Pt NPs then become stable after ~100 PH cycles, even when annealed under isothermal heating at 1000 °C (does this mean isothermal heating can also freeze sintering?).

This experimental data indicate that the stability or long-term anti-sintering of Pt NPs is NOT (at least not entirely) from pulsed heating effect, but rather, and importantly, from the strong interaction with the substrate. This conclusion is understandable and convincing, but it is in strong contradiction to the authors' claim that pulsed heating freezes sintering in supported metal nanoparticles.

A clearer separation of the early-stage kinetic suppression of sintering and the late-stage chemical or interfacial stabilization, along with a discussion of the exact role of pulsed heating, would substantially strengthen the mechanistic clarity of the work.

It seems that, compared with slow continuous heating, PH is effective in building strong metal-support interaction at the initial stage by competing with sintering and graphitization behaviors. But, the long-term stability of nanoparticles under PH is not because of PH freezing, but because of the formation of strong interfacial bonding. The authors should either prove experimentally or theoretically that pulsed heating can indeed freeze sintering, or attribute the true role of PH to first forming the strong bonding, and then ensuring stability.

Overall, the discussion of beyond thermodynamics, pulsed heating freezes sintering, the thermodynamically unstable yet kinetically stable state under PH, etc, lacks scientific rigor and experimental/theoretical proof. The authors should elaborate the paper strictly based on their observations.

Another one, why are responses and data in comment #3 not revealed in either the main text or SI? The discussion of possible kinetic competition among crystallization, bond formation, sintering, and graphitization is helpful for readers to understand the role and design of PH.

Point-by-point response

We sincerely thank the reviewers for their comments and suggestions on our manuscript. We have addressed all the concerns raised by the referees in full and revised the manuscript accordingly. **Our responses** to the reviewers' comments are given below.

The content is organized as following:

Section I: Response for Reviewer 1 (p2-p16)

Section II: Response for Reviewer 2 (p17-p25)

Response for Reviewer #1

In the manuscript, Huang et al. report ultrafast pulsed heating as a nonequilibrium strategy to suppress the sintering of supported Pt nanoparticles, using graphene as the support. Through a combination of in situ STEM analysis, EELS, DFT, and MD simulations, the authors reveal that pulsed heating induces a thermodynamically unstable yet kinetically stable state, enabling improved crystallinity and enhanced metal–support interactions while effectively inhibiting particle migration and coalescence. This work provides a novel atomic-scale understanding of metastable states in nanocatalysts and offers an alternative route to sintering-resistant catalyst synthesis beyond thermodynamic constraints. Overall, the authors present a solid and well-validated study, and this manuscript will be of high interest to researchers working on nanocatalysts. The methodology is rigorous, and the mechanistic insights are valuable. This manuscript can be considered for publication in Nature Communications. However, the following points should be addressed before publication to further improve the quality of this work.

(1) The authors mention that the heating pattern was designed based on recent reports; however, a more detailed description of the temperature profile, particularly the cooling rate following each pulsed heating, is necessary. Since cooling kinetics are critical for the formation of metastable nanostructures, quantifying the cooling behavior would greatly enhance the mechanistic understanding. Additionally, the selection of the 10-pulse condition as the primary reference over the 1-pulse case remains unclear. Could the authors clarify and comment on the potential effects of a single, higher-temperature pulse?

Our response: Thanks very much for the positive comment and kind suggestion from the reviewer. We would like to respond for your concerns as follows.

1) Regarding the reviewer's comments on the temperature profile, we emphasize that both the heating rate and, in particular, the cooling rate of 150 °C/s in our pulsed heating experiment are well within the capabilities of the DENSSolutions Wildfire heating

system.^{1,2} Furthermore, direct temperature measurements recorded by the Impulse software during an individual heating pulse confirm the uniformity of the designed heating/cooling rate (**Table R1**).

Second, we agree with the reviewer that the cooling kinetics indeed play a decisive role in the formation of metastable structures. The cooling rate achieved in our pulsed heating experiments is markedly higher than that of conventional heating, enabling the suppression of atomic diffusion and thereby stabilizing the metastable structures formed at elevated temperatures.^{3,4} In contrast, conventional cooling rates keep the system much close to thermodynamic equilibrium, which facilitates the crystallization of metal nanoparticles and the evolution of thermodynamically stable structures. Consequently, the cooling behaviors of the two approaches differ fundamentally. To further address this point, we have now included detailed descriptions of the cooling rate in the revised text, which read as “Particularly, the design of ultrafast cooling rates is essential, as the quenching process is critical for stabilizing the metastable structures formed at high temperatures.^{45,46}” and “For comparison, another in situ experiment under conventional heating was parallelly conducted, involving a slow heating rate of 1 °C s⁻¹ to 1000 °C and a subsequent cooling rate of 5 °C s⁻¹ to room temperature”.

2) Regarding the reviewer’s questions on the design of pulse conditions, we note that the choice of a 10-pulse sequence follows the approach reported by Yao et al.;⁵ since both the decomposition of metal precursors and the nucleation/growth of metal NPs require sufficient time, and a single heating pulse is typically inadequate to complete these processes. Approximately 10 pulses are therefore necessary to obtain uniformly dispersed NPs, which then provides a meaningful basis for investigating the sintering behavior of supported nanocatalysts.

Furthermore, while a single, higher-temperature pulse can indeed accelerate precursor decomposition, organics volatilization, and NP crystallization,⁶ it does not alter the intrinsic tendency for sintering or structural evolution under pulsed heating; rather, it merely accelerates the kinetics. Thus, we consider higher-temperature pulses with fewer

pulse numbers cannot be the ideal parameter. We also want to remind you that our focus lies on the understanding of sintering dynamics, structural evolution of NPs, and interface optimization that occur during pulsed heating; therefore, employing relatively lower-temperature pulses slows down these transformations, thereby enabling more convenient observations.

To clarify the reason for the design in our work, we have now added the following description in the updated text: “the heating pattern was set based on recent reports and the initial 10-pulse condition was selected to complete the decomposition of precursors and capture the crystallization process of metal NPs”.

References:

- (1) van Swieten, T. P. *et al.* Mapping elevated temperatures with a micrometer resolution using the luminescence of chemically stable upconversion nanoparticles. *ACS Appl. Nano Mater.* **4**, 4208-4215 (2021).
- (2) Vijayan, S., Wang, R., Kong, Z. & Jinschek, J. R. Quantification of extreme thermal gradients during in situ transmission electron microscope heating experiments. *Microsc. Res. Techniq.* **85**, 1527-1537 (2021).
- (3) Han, Y.-C., Cao, P.-Y. & Tian, Z.-Q. Controllable synthesis of solid catalysts by high-temperature pulse. *Acc. Mater. Res.* **4**, 648-654 (2023).
- (4) Deng, B. *et al.* Phase controlled synthesis of transition metal carbide nanocrystals by ultrafast flash Joule heating. *Nat. Commun.* **13**, 262 (2022).
- (5) Yao, Y. *et al.* High temperature shockwave stabilized single atoms. *Nat. Nanotechnol.* **14**, 851-857 (2019).
- (6) Chen, L., Liu, J., Volpe, R. & Grasso, S. Impact of repeated heat pulses on the consolidation and grain growth of 3YSZ consolidated via ultrafast high-temperature sintering (UHS). *J. Eur. Ceram. Soc.* **45**, 117395 (2025).

Table R1. The actual temperature measurement of a single heating pulse.

Experiment time / s	Temperature Setpoint / °C	Temperature Measured / °C
0	20	20.507456
1.982	347.413451	305.582642
3.965	628.539899	651.154297
6.610	1003.617347	1012.758667
8.591	768.879477	800.145874
10.571	487.144688	474.15567
13.542	20	20.297533

(2) The authors mention that ultrafast pulsed heating effectively suppresses atomic diffusion and aggregation of nanoparticles. However, in several recent studies, including direct-joule heating-based redispersion/atomization (Journal of the American Chemical Society 142,41, 2020, 17364-17371., Science 359, 6383, 2018, 1489-1494.), and photothermal effect-driven redispersion/atomization (metal oxide redispersion (Advanced Materials, 2025, 2419790.), Pt atomization (Chem 8, 4, 2022, 1014-1033.)), it has been shown that ultrafast high-temperature treatment can actually promote atomic diffusion while simultaneously restricting aggregation, resulting in smaller or even atomically dispersed particles. Could the pulsed heating strategy used in this study also induce redispersion or partial atomization of Pt nanoparticles, especially at elevated pulse temperatures or with increased pulse counts? In such cases, atomic mobility could be activated while aggregation remains kinetically suppressed due to short exposure durations. The authors are encouraged to comment on whether such size reduction or dispersion behavior could emerge under more intensive pulse conditions and whether this pathway has been experimentally tested or theoretically predicted.

Our response: We sincerely thank the reviewer for this important suggestion. We clarify that no redispersion or atomization behavior has been observed in our pulsed heating experiments, as evidenced by the size evolution of Pt NPs over 100 heating pulses (**Figure 3h**, shown here as **Figure R1**).

One possible reason is the size effect: pulsed heating decomposes the precursors into well-dispersed NPs with diameters below 2 nm, which are much smaller than the redispersed NPs reported in the literature cited by the reviewer. Such ultra-small NPs possess very high surface energies, making further redispersion unfavorable. More importantly, as noted in previous reports,⁷⁻⁹ redispersion or atomization typically requires two key conditions: (i) sufficiently high temperatures to break atomic bonds within the NPs and overcome diffusion barriers, thus causing the NPs to split and move on the support; (ii) strong metal–support interactions, such as spatial (MOFs, COFs) or chemical (surface functional groups, dangling bonds) confinements, are also essential to stabilize the redistributed atoms or ultra-small clusters.

Therefore, in our opinion (and under the current experimental conditions), redispersion/atomization cannot be induced solely by elevated pulse temperatures or increased pulse counts because the nature of the support is also critical. In our system, the graphene support provides an atomically smooth surface with negligible confinement, preventing particles/atoms trapping. We emphasize it here that this design is intentional, as it allows us to focus on the sintering behaviors of NPs under pulsed heating.

References:

- (7) Lang, Z. et al. Destabilization of single-atom catalysts: characterization, mechanisms, and regeneration strategies. *Adv. Mater.* **37**, 2418942 (2025).
- (8) Li, L. & Zhang, N. Atomic dispersion of bulk/nano metals to atomic-sites catalysts and their application in thermal catalysis. *Nano Res.* **16**, 6380-6401 (2023).
- (9) Morgan, K., Goguet, A. & Hardacre, C. Metal redispersion strategies for recycling of supported metal catalysts: a perspective. *ACS Catal.* **5**, 3430-3445 (2015).

Figure R1. (Figure 3h) Average size of Pt NPs as a function of pulse number across (a-f). Error band represents the 95% confidence interval of the mean.

(3) The manuscript reports the application of up to 100 successive heating pulses, yet no notable structural or chemical change in the graphene support is discussed. Could the authors elaborate on the methods used to evaluate the integrity of the graphene after repeated thermal treatments? In particular, were any atomic-level characterizations (e.g., Raman spectroscopy, EELS, or aberration-corrected TEM) conducted to assess possible cumulative damage or defect generation?

Our response: We thank the reviewer for this professional suggestion. While we did not initially emphasize the changes in the graphene support during 100 heating pulses, it is well established that graphene possesses excellent thermal stability as a catalyst support, as frequently documented in previous studies.^{10,11} Additionally, there are also studies indicating that surface defects generated during preparation can be repaired under prolonged heating,¹² but the extremely short duration of each pulse in our experiments makes it unlikely that successive pulses would significantly affect the structure of graphene. Moreover, the graphene samples were sintered at 500 °C for 30 minutes prior to our in situ heating experiments, which further ensures their thermal stability.

To directly assess potential cumulative damage or defect generation of graphene support, we collected the EELS spectra of pristine graphene before and after 100

heating pulses (**Figure R2**). The absence of noticeable changes in the C–K edge confirms that pulsed heating exerts negligible influence on the graphene structure. We have now added the corresponding discussion on the integrity of graphene under repeated pulses in the revised manuscript, which reads as “**In addition, the C K-edge EELS results of the graphene support before and after 100 heating pulses confirm its structural integrity under repeated heating pulses (Figure S13).**”. We also added **Figure R2** in the updated Supplementary Information as **Figure S13**.

References:

- (10) Jang, I. et al. Instantaneous thermal energy for swift synthesis of single-atom catalysts for unparalleled performance in metal–air batteries and fuel cells. *Adv. Mater.* **36**, 2403273 (2024).
- (11) Qiao, Y. et al. 3D-printed graphene oxide framework with thermal shock synthesized nanoparticles for Li-CO₂ batteries. *Adv. Funct. Mater.* **28**, 1805899 (2018).
- (12) Liu, L., Qing, M., Wang, Y. & Chen, S. Defects in graphene: generation, healing, and their effects on the properties of graphene: a review. *J. Mater. Sci. Technol.* **31**, 599-606 (2015).

Figure R2. (Figure S13) (a, b) STEM image and C K-edge EELS spectra of graphene support before and after 100 heating pulses.

(4) The discussion compares the appearance of the Pt (111) facet after a single pulsed heating step with the delayed emergence of the same facet under conventional heating (initially showing the (200) facet). Given that (111) is thermodynamically more stable than (200), it would be more accurate to highlight the rapid stabilization of (111) as a

merit of pulsed sintering rather than concluding a higher degree of crystallinity. Moreover, as Figure S12 also shows the appearance of the (200) facet under pulsed heating, please address this apparent contradiction.

Our response: We appreciate the reviewer's insightful comment on that pulsed heating promotes the rapid formation of the thermodynamically stable (111) facet. However, the conclusion we drew in the text regarding higher crystallinity was not directly linked to the emergence of the (111) plane. Instead, it was based on the brightness contrast of the diffraction spots in the FFT patterns (**Figure S9** and **Figure 2m**, attached here as **Figure R3** and **Figure R4**, respectively), where pulsed heating produced stronger diffraction contrasts, qualitatively indicating a higher degree of NP crystallinity.¹³ To avoid potential misunderstanding caused by our earlier wording, we have revised the corresponding statement in the manuscript as: "In contrast, the exposed crystal plane for Pt NPs under the conventional heating process changed from (200) to (111) after 500 °C (**Figure 2l**) and these NPs displayed poor crystallinity (see the corresponding FFT pattern in **Figure S9b**);".

Regarding the appearance of the (200) facet after 100 heating pulses, we clarify that while this is possible, it remains a rare phenomenon, likely arising from anchoring effects of surface defects or localized interfacial forces. Importantly, the overall trend is the preferential formation of the more stable (111) facets, as shown in **Figure R5**, where the ratio of (111) facet exceeds 90%. To illustrate this point more clearly, we have replaced the original **Figure S12** with **Figure R5** in the updated Supplementary Information and revised the corresponding text, which read as "high-resolution TEM images of the Pt NPs after 100 pulses in **Figure S14** exhibit hexagonal NPs with dominant {111} facets and occasional (200) planes".

Reference:

(13) Fei, L. et al. Observable two-step nucleation mechanism in solid-state formation of tungsten carbide. *ACS Nano* **13**, 681-688 (2019).

Figure R3. (Figure S9) (a) FFT pattern series corresponding to pulsed heating process in **Figure 2k**. (b) FFT pattern series corresponding to conventional heating process in **Figure 2l**.

Figure R4. (Figure 2m) Evolutions of relative crystalline orders for Pt NPs throughout (k) and (l) against time. The amounts of crystalline order were determined from the spot intensity of the corresponding FFT patterns (see **Figure S9**), comparing the value from a NP with the same crystal orientation (whose crystalline order is set as 100, heated with 10 pulses).

Figure R5. (Figure S14) High-resolution TEM images showing the crystallographic analysis of Pt NPs after 100 heating pulses. NPs that expose their (111) and (200) crystal planes are marked with green and yellow circles, respectively. The result indicates the dominance of the stable (111) crystal plane.

(5) A noticeable shift in the peak position begins at the 4th pulse. Please provide a discussion on this phenomenon. Additionally, prior to comparing 10-pulse treatment with conventional heating for 1000 seconds, consider providing comparative data at intermediate temperatures to better contrast single-pulse effects.

Our response: We sincerely thank the reviewer's constructive comment.

1) The phenomenon of peak shift noticed by the reviewer can be explained by the fact that the first few pulses primarily drive the decomposition of precursor, during which the newly formed NPs are not yet in intimate contact with the support due to residual organics at the interface. By the 4th pulse, the NPs may have established closer contacts with the graphene support, enabling charge transfer, as reflected by the peak shift in the EELS spectra (**Figure 2n**, attached here as **Figure R6**). That charge transfer between graphene and Pt NPs has been confirmed (both experimentally and theoretically) in our previous work.¹⁴ To clarify this point, we have added the corresponding description in the updated text, which read as “As shown in **Figure 2n**, the π^* excitation of C K-edge gradually shifts toward lower energy loss values during pulsed heating (noticeable shift begins at the 4th pulse, likely due to the NPs initiate close contacts with the support), which can be a result of strong bonding and charge transfer between Pt NPs and graphene support that modify the electronic structure around the active sites;”.

2) Regarding the EELS results under conventional heating (**Figure 2n**), we clarify that these measurements were conducted during the heating process rather than under isothermal conditions at 1000 °C. Comparative spectra at intermediate temperatures have therefore been included in **Figure 2n**.

Reference:

(14) Gan, M. et al. Incomplete charge transfer bestows significant sintering resistance for metal nanoparticles on two-dimensional graphyne. *J. Mater. Chem. A* **12**, 29174-29183 (2024).

Figure R6. (Figure 2n) C K-edge EELS spectra for the graphene supports as a function of heating time under pulsed heating and conventional heating modes.

(6) In the interpretation of Figure 3j, clarify the rationale for lattice matching discussion between 3D Pt nanoparticles and 2D graphene sheets. In particular, provides more detail on how the crystallographic orientation was determined from FFT patterns.

Our response: Thanks very much for the constructive comments from the reviewer.

1) For your concern on **Figure 3j**, our intention for this scheme focuses on the crystallographic orientation relationship between the [111] direction of Pt NPs and the [10-10] direction of graphene flakes (you may also refer to **Figure 5c** for the side-view of this interfacial relationship), which has been well explored in previous studies.¹⁵ To

clearly illustrate this orientation relationship and to avoid potential misunderstanding, we have revised the original **Figure 3j** with **Figure R7** in the updated manuscript.

In addition, to clarify the validity of the “lattice matching” statement in the manuscript, we have calculated the lattice mismatch between the Pt (111) plane and the graphene (10-10) plane, which is 6.57%. Since this value exceeds the typical lattice matching criterion (<5%), the original statement of “lattice matching” was indeed inappropriate. We apologize for this oversight, and following the suggestion of Reviewer #2, we have now revised the description of “lattice matching” to “orientational alignment” in the updated text.

2) In response to how the crystallographic orientation was derived from the FFT patterns, we first determined the crystallographic orientation of the graphene flake from the FFT pattern (**Figure 3i**, shown here as **Figure R8**); subsequently, the corresponding crystallographic orientation of Pt NPs supported by the above graphene flake were measured (**Figure R9**). Then the θ angles between the graphene [10-10] direction and the Pt [111] direction were determined to assess the orientation alignment. We found that the distributions of θ angle were mainly around 0° , 60° , 120° , and 180° , indicating that the Pt NPs adopted preferred orientations aligned with the graphene support during the pulsed heating process. To illustrate the acquisition of crystallographic orientation, we have added **Figure R9** in the updated Supplementary Information as **Figure S15** and revised the corresponding text, which reads as “When the Pt NP orientation (green dashed line) deviated from the graphene orientation, the angle was denoted as θ , as shown in **Figure 3j** and **Figure S15**.”.

Reference:

(15) Cai, P.-Y. et al. Atomic structures of Pt nanoclusters supported on graphene grown on Pt(111). *J. Phys. Chem. C* **122**, 28, 16132-16141.

Figure R7. (Figure 3j) Atomic schematic illustrating the orientation of Pt NPs with respect to graphene support.

Figure R8. (Figure 3i) SAED pattern of the graphene support, showing its single-crystalline nature.

Figure R9. (Figure S15) (a, b) HRTEM images and corresponding FFT pattern of Pt NPs supported on graphene support, showing their crystallographic orientation relationships against graphene flakes.

(7) Only four images are shown for the 100-pulse case in Figure 4(e-h). Please revise the layout for consistency across all pulse conditions or clarify the rationale behind the image selection.

Our response: Thank the reviewer very much for this meticulous comment. We clarify that the structural evolution of Pt NPs shown in **Figure 4** (attached here as **Figure R10**) does not represent an in situ sequence. In fact, to ensure experimental reproducibility, we conducted three independent experiments and selected Pt NPs with representative structures at different pulse numbers to illustrate the structural adjustment process. In particular, typical NPs observed along the [01-1] zone axis after 100 heating pulses are presented in **Figure 4e-h**, as this configuration is relatively the most stable and represents the eventual structural tendency of Pt NPs under pulsed heating. To avoid ambiguity, we have revised the description in the updated text, which read as “**In order to directly disclose the structural optimization of Pt NPs, we accordingly captured the high-resolution TEM images of graphene supported Pt NPs with representative structures along with the increasing heating pulses.**”.

Figure R10. (Figure 4) Atomic-scale evolutions of Pt NPs in 100 heating pulses. High-resolution TEM images showing the lattice structure of representative Pt NPs after (a, b) 10 pulses, (c, d) 40 pulses, and (e-h) 100 heating pulses. Scale bar in (a) also applies to (b-h). (i-j) False-colored iFFT patterns showing the atomic arrangements of Pt NPs corresponding to (e-h), respectively. (m) Schematic illustration for the structural evolution of Pt NPs on a graphene support via pulsed heating.

(8) Confirm whether the TEM image in Figure S13 was obtained along the same zone axis as those shown in Figure 4. This is important for accurate structural comparison.

Our response: Thanks very much for the reviewer's reminder. The TEM images in **Figure S13** and **Figure 4** were not acquired along the same zone axis. As indicated in the manuscript, the repeated pulsed heating promotes Pt NPs to adopt a stable structure along the [01-1] zone axis on graphene, whereas conventional heating in our experiments does not allow the NPs to fully reach the most thermodynamically stable state due to the relatively short heating duration. As a result, NPs under conventional heating may form along multiple zone axis, such as the [01-1] and [1-21] zone axis, as shown in **Figure R11**. It should be noted that the crystallographic configuration shown in **Figure S13** is the dominant structure in our TEM observations. For a more accurate structural comparison, we have replaced the original **Figure S13** with **Figure R11** in the updated Supplementary Information.

Figure R11. (Figure S16) HRTEM images and corresponding FFT patterns of Pt NPs along the (a, b) [01-1] and (c, d) [1-21] zone axis after conventional heating.

Response for Reviewer #2

The manuscript titled “Defying thermodynamics: Nonequilibrium pulsed heating freezes sintering of supported metal nanocatalysts” explores the behavior of Pt nanoparticles supported on graphene under ultrafast pulsed heating, using in situ (S)TEM observations in combination with DFT and MD simulations. The study aims to demonstrate that pulsed heating leads to a metastable, kinetically stabilized state that can inhibit nanoparticle sintering more effectively than conventional heating. While the experimental design is technically sound and the integration of microscopy with modeling is methodologically appropriate, the current version of the manuscript raises several concerns that must be addressed before it can be considered for publication. The mechanistic claims, particularly regarding structural optimization, interface effects, and the generality of the proposed anti-sintering mechanism, require clearer justification and more critical discussion. Additionally, the manuscript would benefit from a more rigorous evaluation of the crystallinity data, the role of the substrate, and the absence of catalytic performance validation. Therefore, I recommend major revisions. Specific comments are provided below to help the authors improve the manuscript’s clarity, depth, and scientific rigor.

(1) In Figure 2m, the lower crystallinity of Pt NPs under conventional heating may be biased by the different thermal states during which the measurements were taken—if FFT was performed at high temperature, thermal vibrations and atomic diffusion could artificially reduce apparent crystallinity. In contrast, pulsed heating benefits from repeated cooling, effectively annealing the structure. A fair comparison should assess crystallinity after cooling in both cases, or this limitation should be clearly discussed to avoid misinterpretation. Including such a control or at least discussing this thermal-state-dependent artifact would significantly enhance the credibility of the crystallinity comparison and deepen our understanding of the thermal evolution of NP structure in both regimes.

Our response: We thanks very much for the positive comment and kind suggestion

from the reviewer. Following the reviewer's suggestion, we have further compared the crystallinity of the Pt NPs after cooling under both pulsed and conventional heating. The HRTEM image and corresponding FFT pattern of the same Pt NP after conventional heating and cooling are shown in **Figure R12a**, and the relative crystallinity of Pt NPs after cooling is added to the original **Figure 2m**, which is shown here as **Figure R12b**. As expected, high temperatures reduce the crystallinity of Pt NPs, as indicated by the lower FFT intensity for samples heated to 1000 °C under conventional heating (**Figure R12b**). After cooling, the crystallinity under conventional heating increases slightly but remains lower than that observed under pulsed heating. This result suggests that pulsed heating does improve the crystallinity of the NPs, possibly due to its merit of repeated cooling process.

To better illustrate this point, the results (**Figure R12**) have been added to the updated Supplementary Information as **Figure S10**, and the corresponding discussion has been revised in the manuscript, which reads as “(the comparison after cooling is also provided in **Figure S10** to exclude the temperature difference)”.

Figure R12. (Figure S10) (a) HRTEM image and corresponding FFT pattern of the same Pt NP in conventional heating (**Figure 2l**) after cooling to RT. (b) Evolutions of relative crystalline orders for Pt NPs throughout the pulsed heating and conventional heating, including the crystallinity comparison after cooling.

(2) In Figure 2n, the electron energy loss spectroscopy (EELS) results suggest changes at the metal-support interface, but the chemical state evolution of Pt (e.g., oxidation state, bonding environment) is not fully explored. Complementary characterization

would help clarify the electronic effects contributing to NP stabilization.

Our response: Thank the reviewer very much for this important suggestion. We attempted to probe the chemical state of Pt using EELS; however, due to the relatively high energy-loss of Pt and the considerable thickness, the signal was too weak to reliably determine the chemical state of Pt, as shown in **Figure R13**.

Instead, we examined the chemical state evolution of Pt indirectly through the changes in the C K-edge EELS spectra. First, EELS spectra of pristine graphene before and after 100 heating pulses were collected (**Figure R14**), showing no significant changes. In contrast, a notable C K-edge shift was observed at the Pt NP-graphene interface (**Figure 2n**), suggesting that this shift originates from charge transfer between Pt NPs and graphene rather than from the graphene itself. In addition, our previous work has demonstrated that charge transfer generally occurs from Pt NPs to graphene, resulting in a decrease in the C valence and a corresponding increase in the Pt valence.¹⁶ Accordingly, this charge transfer during pulsed heating leads to partial oxidation of Pt and the formation of strong Pt-C bonds, which contribute to NP stabilization.

We have revised the corresponding discussion in the revised manuscript, which reads as “As shown in **Figure 2n**, the π^* excitation of C K-edge gradually shifts toward lower energy loss values during pulsed heating (noticeable shift begins at the 4th pulse, likely due to the NPs initiate close contacts with the support), which can be a result of strong bonding and charge transfer between Pt NPs and graphene support that modify the electronic structure around the active sites;” and “In addition, the C K-edge EELS results of the graphene support before and after 100 heating pulses confirm its structural integrity under repeated heating pulses (**Figure S13**).”. We also added **Figure R14** in the updated Supplementary Information as **Figure S13**.

Reference:

(16) Gan, M. et al. Incomplete charge transfer bestows significant sintering resistance for metal nanoparticles on two-dimensional graphyne. *J. Mater. Chem. A* **12**, 29174-

29183 (2024).

Figure R13. EELS spectra taken from a Pt NP after pulsed heating for the Pt-M₅ edge.

Figure R14. (Figure S13) (a, b) STEM image and C K-edge EELS spectra of graphene support before and after 100 heating pulses.

(3) In Figure 3, the manuscript claims a “high degree of lattice matching” between Pt NPs and graphene supports. However, this terminology is not strictly accurate. Pt(111) has a lattice spacing of ~ 0.227 nm, while the in-plane lattice spacing of graphene (e.g., (10-10)) is ~ 0.213 nm, and their crystallographic symmetries (fcc vs. hexagonal) differ. Thus, this interface should not be described as lattice-matched. A more appropriate term would be “orientation alignment” or “epitaxial-like interfacial relationship.” The authors are advised to revise this statement for precision.

Our response: Thank the reviewer very much for this significant suggestion. To clarify the validity of the “lattice matching” statement in the manuscript, we have calculated the lattice mismatch between the Pt (111) plane and the graphene (10-10) plane, which

is 6.57%. Since this value exceeds the typical lattice matching criterion (<5%), the original statement of “lattice matching” was indeed inappropriate. We apologize for this oversight, and following your suggestion, we have now revised the description of “lattice matching” to “orientational alignment” in the updated text.

(4) In the part of “Sintering kinetics of pulsed heating”, The manuscript attributes the sintering resistance under pulsed heating to both interfacial structural optimization and kinetic suppression. It would strengthen the mechanistic understanding if the authors discuss whether kinetic limitation alone—without crystallographic alignment or structural optimization—would be sufficient to ensure long-term stability. This distinction is important to evaluate whether the observed robustness is primarily due to the heating profile or also critically dependent on support-induced ordering.

Our response: Thanks very much for the reviewer’s reminder.

First, in the supported catalyst systems without noticeable interfacial interaction, pulsed heating alone can still effectively limit sintering,^{17,18} suggesting that pure kinetic confinement is sufficient to inhibit particle migration. Similarly, the Pt/graphene system used in our study exhibits weak metal-support interactions.¹⁹ In particular, during the first few heating pulses, when NPs have not yet established good contacts with graphene support, sintering is thoroughly suppressed by kinetic blocking of atomic diffusion and particle migration. Therefore, we conclude that even under a single pulse, or cumulatively across 100 pulses, kinetics alone can prevent sintering.

Second, repeated pulses lead to structural and interfacial optimization that enhances metal-support interactions, further contributing to sintering resistance. Nonetheless, kinetic suppression remains the dominant factor. We have incorporated this discussion in the revised manuscript, which reads as “and second, the NPs are kinetically hindered from undergoing effective migration due to the brief duration of heating pulses, which is the key mechanism underlying their sintering resistance.”.

References:

(17) Yao, Y. et al. In situ high temperature synthesis of single-component metallic nanoparticles. *ACS Cent. Sci.* **3**, 294-301 (2017).

(18) Yao, Y. et al. High temperature shockwave stabilized single atoms. *Nat. Nanotechnol.* **14**, 851-857 (2019).

(19) Gan, M. et al. Incomplete charge transfer bestows significant sintering resistance for metal nanoparticles on two-dimensional graphyne. *J. Mater. Chem. A* **12**, 29174-29183 (2024).

(5). Although 100 pulses demonstrate improved sintering resistance, the net thermal exposure remains short relative to real catalytic conditions. The authors are encouraged to address how these metastable structures behave under extended heating, cyclic heating, or in the presence of reactive environments, either through additional data, cited references, or detailed discussion.

Our response: Thanks very much for the reviewer's reminder. First, numerous studies have shown that nanocatalysts synthesized via pulsed heating exhibit excellent catalytic stability under prolonged heating or reactive environments,²⁰⁻²² which have been attributed to the formation of metastable structures and/or the strong metal-support interactions. Second, to directly assess the thermal stability of the Pt/graphene nanocatalyst formed after 100 heating pulses in our work, we conducted an extended heating test by maintaining the metastable nanostructure at 1000 °C for 1 hour. As shown in **Figure R15**, the Pt NPs retained excellent thermal stability with no obvious sintering. This stability can be attributed to the strong bonding between Pt nanostructures and the graphene support, which arises from the combined effects of kinetic suppression and interface optimization during pulsed heating, effectively preventing NP sintering even under prolonged high-temperature exposure.

We have now added the result (**Figure R15**) to the updated Supplementary Information as **Figure S18** and integrated this discussion in the revised manuscript, which reads as "In addition, to assess the thermal stability of the Pt/graphene nanocatalyst formed after 100 heating pulses, we conducted an extended heating test by maintaining it at 1000 °C

for 1 hour. As shown in **Figure S18**, the Pt NPs retained excellent thermal stability with no obvious sintering.”.

References:

(20) Yao, Y. et al. High temperature shockwave stabilized single atoms. *Nat. Nanotechnol.* **14**, 851-857 (2019).

(21) Li, C. et al. Ultrafast self-heating synthesis of robust heterogeneous nanocarbides for high current density hydrogen evolution reaction. *Nat. Commun.* **13**, 3338 (2022).

(22) Du, P. et al. In-situ Joule-heating drives rapid and on-demand catalytic VOCs removal with ultralow energy consumption. *Nano Energy* **102**, 107725 (2022).

Figure R15. (Figure S18) In situ STEM images showing the time-lapsed thermal stability of Pt/graphene nanocatalyst formed after 100 heating pulses at 1000 °C.

(6) While structural stabilization is clearly demonstrated, the manuscript lacks direct correlation with catalytic performance. A brief experimental validation (e.g., a model reaction demonstrating enhanced durability or activity of pulsed-heated Pt NPs) or a robust discussion on how the observed microstructural advantages may translate into catalytic improvements would significantly enhance the relevance of the study.

Our response: We sincerely thank the reviewer for this constructive comment. As we addressed in the response above, the Pt/graphene nanocatalyst formed by 100 heating pulses in our work exhibits great thermal stability under extended heating, as shown in **Figure R15**, which is crucial for practical catalytic performance.

Then, we will clarify the relationship between the microstructural advantages resulting from pulsed heating and the catalytic performance in terms of size, structure and metal-support interaction for nanocatalysts.

First, the ultrafast heating/cooling kinetics of pulsed heating limit the sintering effect, producing ultra-small metal NPs (<3 nm) with high dispersibility. This results in a large specific surface area and abundant active sites, directly enhancing catalytic activity.^{23,24}

Second, the high crystallinity of NPs achieved under pulsed heating, particularly the metastable structure arising from multistep optimization, reduces lattice defects (and inactive sites) while increasing resistance to atomic migration and sintering.²⁵⁻²⁷ This contributes to higher intrinsic activity and provides a structural basis for long-term stability.

Third, strong electronic metal-support interaction (ESMI), induced by charge transfer during pulsed heating, effectively anchors the metal NPs, suppressing migration, agglomeration, and Ostwald ripening during catalysis.^{28,29} This mechanism underlies the minimization of performance degradation over prolonged operation. In addition, ESMI can modulate the electronic structure of the metal surface, optimize the adsorption strength of the reactants/intermediates and potentially influence the reaction path.³⁰

Therefore, we have now added the corresponding discussion on how the observed microstructural advantages translate into catalytic improvements in the updated text, which reads as “On the basis of above analysis, we want to finally discuss the microstructural advantages of those pulsed heating-derived NPs in catalytic applications: 1) the ultra-small (<3 nm) and highly dispersed NPs provide abundant

active sites, which are promising for improving the mass-specific catalytic activity;^{22,47} 2) the high-crystalline NPs together with the metastable metal/support structure ensure high intrinsic activity as well as excellent sintering resistance;^{49,58,59} and 3) the consequential metal-support interaction (i.e., charge transfer) enhances substantial interfacial stability and modulates the electronic structure of metal NPs, which are beneficial for long-term catalytic stability and reaction selectivity.^{53,54,60} The synergistic effect of these structural advantages directly underpins the catalytic performance and long-term stability of these catalysts, demonstrating the great potential of pulsed heating for precise and facile regulations of supported nanocatalysts.”.

References:

- (23) Dai, Y., Lu, P., Cao, Z., Campbell, C. T. & Xia, Y. The physical chemistry and materials science behind sinter-resistant catalysts. *Chem. Soc. Rev.* **47**, 4314-4331 (2018).
- (24) Liu, L. & Corma, A. Metal catalysts for heterogeneous catalysis: from single atoms to nanoclusters and nanoparticles. *Chem. Rev.* **118**, 4981-5079 (2018).
- (25) Dou, X. et al. Structure–reactivity relationship of zeolite-confined Rh catalysts for hydroformylation of linear α -olefins. *J. Am. Chem. Soc.* **147**, 2726-2736 (2025).
- (26) Chen, Q. et al. Rapid synthesis of metastable materials for electrocatalysis. *Chem. Soc. Rev.* **54**, 4567-4616 (2025).
- (27) Li, Z. et al. Well-defined materials for heterogeneous catalysis: from nanoparticles to isolated single-atom sites. *Chem. Rev.* **120**, 623-682 (2020).
- (28) Xu, M. et al. Renaissance of strong metal–support interactions. *J. Am. Chem. Soc.* **146**, 2290-2307 (2024).
- (29) Hu, X., Xu, D. & Jiang, J. Strong metal - support interaction between Pt and TiO₂ over high-temperature CO₂ hydrogenation. *Angew. Chem. Int. Ed.* **64**, e202419103 (2025).
- (30) Li, J. et al. Highly active and stable metal single-atom catalysts achieved by strong electronic metal–support interactions. *J. Am. Chem. Soc.* **141**, 14515-14519 (2019).

Point-by-point response

We sincerely thank the reviewers for their comments and suggestions on our manuscript. We have addressed all the concerns raised by the referees in full and revised the manuscript accordingly. **Our responses** to the reviewers' comments are given below.

The content is organized as following:

Section I: Response for Reviewer 1 (p2)

Section II: Response for Reviewer 2 (p3-p9)

Response for Reviewer #1

This study played a crucial role in elucidating the mechanism for rational nanostructured catalyst design through heat treatment in a metastable state. Furthermore, I believe it is fully acceptable for accurately answering the questions posed.

Our response: We sincerely appreciate the reviewer's positive comment and recommendation for publication.

Response for Reviewer #2

Many concerns have been addressed. However, after careful review of the revised manuscript and rebuttal, there are still major questions for this paper.

(1) What dose the authors mean about “defying thermodynamics”. How did you achieve that and how can anyone defy thermodynamics? A mere anti-sintering behavior is not defying thermodynamics.

Our response: We thank the reviewer for this careful and important question. We fully agree that no process can literally “violate” the thermodynamic laws, and we regret any ambiguity caused by the phrasing. By using “defying thermodynamics”, we intended to describe a process in which pulsed heating drives the system along a nonequilibrium, kinetically constrained pathway that is inaccessible under conventional, near-equilibrium heating. In other words, pulsed heating does not break thermodynamic laws; instead, it changes the accessible energetic landscape by drastically limiting the time available for diffusion and equilibration, thereby trapping the system in metastable configurations rather than allowing it to evolve toward the global thermodynamic minimum.

In our experiments, this nonequilibrium pathway is realized through three interrelated effects, all directly supported by the *in situ* observations: i) Ultrafast heating and cooling (up to 150 °C s⁻¹) suppress long-range atomic diffusion and particle migration, preventing the system from thermodynamically favorable sintering. ii) Repeated short pulses kinetically freeze intermediate configurations and then allow stepwise structural relaxation and interface optimization rather than full equilibration, as evidenced by the evolution of NP size (**Figure 3h**), the enhanced FFT diffraction-spot contrast (indicating improved crystallinity) (**Figure 2m**), and the EELS C K-edge shifts consistent with charge transfer and strengthened metal–support interactions (**Figure 2n**). iii) The resulting nanostructures are stabilized by improved crystallinity and stronger interfacial bonding; consequently, they exhibit remarkable resistance to coarsening even during extended continuous heating at 1000 °C for 1 h (**Figure S18**).

Thus, our work goes beyond a simple “anti-sintering” phenomenon. Pulsed heating not only suppresses thermodynamically driven coarsening, but also routes nanoparticles along a distinct formation pathway that yields robust structures with specific crystallography and interfacial chemistry thought a thermodynamically unstable yet kinetically stable state (that is difficult for conventional heating to realize). This ability to access useful nonequilibrium configurations by kinetic control is what we intended to convey by the phrase “defying thermodynamics,” not a literal violation of thermodynamic principles. To avoid the possible ambiguity, we have changed this wording in the revised manuscript as “**Beyond thermodynamics**”.

(2) The particle formation and migration process have been previously studied by Nat. Commun. 2020, 11 (1), 6373, and others.

Our response: We thank the reviewer for pointing out the relevant literature. Regarding the Nat. Commun. 2020 paper, it reports fundamentally different experimental facts and mechanisms as compared with this work. In that paper, a single, high-energy pulse is used to modify the support (for example, transforming amorphous carbon into highly defective turbostratic graphite), thereby providing nucleation and anchoring sites for nanoparticles. This process therefore depends on support modification as the stabilization route.

In contrast, our study establishes a particle-centred, kinetic-controlled strategy that does not rely on support transformation. Through repeated short pulses, we kinetically suppress diffusion-driven sintering and allow the metal particles themselves to undergo multistep structural evolution (including improved crystallinity, facet selection, and progressive strengthening of the metal-support interface). This dynamic optimization process yields highly robust supported nanocatalysts even on atomically smooth graphene.

This distinction, between support modification and intrinsic nanoparticle kinetic

optimization, represents the key conceptual advance of our work. It is further substantiated by our *in situ* TEM evidence (particle-size evolution, FFT contrast enhancement, and C K-edge shifts) and by the extended isothermal stability test.

We have cited this paper (*Nat. Commun.* 2020, 11 (1), 6373) and clarified this mechanistic difference in the revised manuscript, which now reads as follows: “It also demonstrates a distinct, particle-centred, and kinetic-controlled pathway in which pulsed heating drives stepwise structural optimization of NPs, in contrast to previous reports where a single high-energy pulse induces support reconstruction and defect generation to stabilize NPs.³⁵”

(3) What is the safe region for ultrafast heating to maintain particle stability, for example, T range, t or pulse times range, and heating/cooling ranges. The authors should dig into the mechanism of this kinetic stabilization and propose design principles, instead of just listing one example.

Our response: We sincerely thank the reviewer for this insightful comment.

First, we wish to emphasize that the central aim of our study is indeed to elucidate the stabilization mechanism of supported metal nanoparticles under pulsed heating. Our findings demonstrate that the pulsed-heating process enables the formation of robust supported NPs through kinetic control—that is, by confining atomic diffusion and structural evolution within a non-equilibrium regime. This kinetic stabilization mechanism remains generally applicable across a broad range of pulsed-heating conditions and is not highly sensitive to minor variations in individual parameters such as pulse duration or cycle number. The experimental parameters selected in this work therefore represent a successful realization within this kinetically accessible regime.

Second, regarding the reviewer’s request for the “safe region” of ultrafast heating, we note that it cannot be defined by a universal, fixed temperature or time range. Instead, the stability regime should be viewed as a “kinetic window” jointly determined by the

interplay among heating/cooling rates, pulse duration, and peak temperature. Besides, the exact boundaries of this “window” depend on the physicochemical properties of the metal-support system (e.g., melting point, interfacial adhesion energy, atomic diffusivity) and the specific experimental environment.

Nevertheless, our *in situ* TEM results offer clear mechanistic insights that can guide the rational design of pulsed-heating protocols. Specifically, nanoparticle stabilization can be achieved when: (i) the heating rate is sufficiently rapid to access a non-equilibrium regime before extensive atomic diffusion occurs; (ii) the pulse duration is short enough to prevent long-range sintering; and (iii) the temperature is high enough to activate local atomic rearrangements that enhance crystallinity and metal–support bonding. Together, these criteria define a broadly applicable kinetic-controlled principle for constructing stable nanocatalysts through pulsed or otherwise temporally modulated thermal stimuli. We believe this provides both a mechanistic foundation and a conceptual framework for extending the strategy to diverse material systems.

Accordingly, we have added the corresponding discussion in the revised manuscript, which now reads as “Consequently, our work offers clear mechanistic insights that NP stabilization for a specific metal during pulsed heating can be achieved within a kinetic window (which should be jointly governed by heating/cooling rates, pulse duration, and peak temperature); under such conditions, diffusion-driven sintering is kinetically confined while local structural rearrangements are activated, enabling the formation of thermally robust nanostructures.”

(4) The authors used stability under pulsed heating as improved stability and sintering-resistance. This is far from enough as thermochemical reactions occur at continuous high T rather than pulses. The true isothermal stability is therefore important to evaluate if the particles are stable or not.

Our response: We fully agree that isothermal stability under continuous high

temperature is essential for practical catalysis. To address this, we have performed an extended isothermal test on the Pt/graphene sample prepared after 100 pulsed-heating cycles by holding it at 1000 °C for 1 hour. As shown in the supported data (**Figure S18**), the Pt NPs show no obvious coarsening or sintering after this prolonged exposure, indicating that the pulsed-heating protocol produces nanostructures that are not only transiently protected by kinetic effects but are also thermally robust under continuous high-T conditions. We interpret this robustness as arising from the combination of improved crystallinity and strengthened metal-support interactions produced during pulsed heating: these features stabilize the kinetically accessed nanostructures against further coarsening even under extended isothermal annealing.

We have added the extended-heating data to the Supplementary Information (**Figure S18**, also shown here as **Figure R1**) and discussed the result in the main text to make this point explicit, which reads as “In addition, to assess the thermal stability of the Pt/graphene nanocatalyst formed after 100 heating pulses, we conducted an extended heating test by maintaining it at 1000 °C for 1 hour. As shown in **Figure S18**, no obvious coarsening or change in particle morphology occurs within the experiment, demonstrating that the produced nanostructures are excellent robust even under continuous high-temperature exposure.”

Figure R1. (Figure S18) In situ STEM images showing the time-lapsed thermal stability of Pt/graphene nanocatalyst formed after 100 heating pulses at 1000 °C.

(5). The conclusion—that pulsed heating achieves a higher degree of crystallinity than conventional heating—should be discussed more cautiously. A conventional heating and a slow cooling provide sufficient time for atoms to diffuse and rearrange into their lowest-energy, most stable lattice positions, thereby minimizing defects and yielding a highly ordered structure. This process (i.e., annealing) is the widely accepted standard method for improving the crystallinity of materials.

Our response: We thank the reviewer for this constructive comment. There might be some misunderstanding caused by our previous wording. Our intention was not to claim that pulsed heating yields a crystallinity higher than that achieved through conventional long-term annealing with slow cooling. Rather, our conclusion highlights that under repeated ultrafast heating pulses, the crystallinity of nanoparticles can be progressively optimized through a unique, kinetically governed pathway. Each heating pulse allows partial structural relaxation and defect healing without enabling extensive diffusion-

driven coarsening, leading to a stepwise improvement of crystalline order. This process differs fundamentally from conventional equilibrium annealing and reflects a nonequilibrium structural optimization mechanism.

To avoid ambiguity, we have revised the relevant statement in the manuscript to emphasize this kinetic improvement of crystallinity, rather than a direct superiority over conventional annealing, which now reads as “As a result, repeated heating pulses drives progressive improvement in the crystallization degrees of Pt NPs and promotes their interfacial optimization with graphene support, leading to structurally refined and thermally robust supported nanocatalysts.”

Point-by-point response

We sincerely thank the reviewers for their comments and suggestions on our manuscript. We have addressed all the concerns raised by the referees in full and revised the manuscript accordingly. **Our responses** to the reviewers' comments are given below.

The content is organized as following:

Section I: Response for Reviewer 1 (p2)

Section II: Response for Reviewer 2 (p3-p14)

Response for Reviewer #1

This study played a crucial role in elucidating the mechanism for rational nanostructured catalyst design through heat treatment in a metastable state. Furthermore, I believe it is fully acceptable for accurately answering the questions posed.

Our response: We sincerely appreciate the reviewer's positive recommendation for our manuscript.

Response for Reviewer #2

The authors did not really address the previous concerns. As shockwave heating to stabilize nanoparticles or nano clusters has been well documented before, using the exact shock method. Now the authors re-performed the experiments under TEM, which, to some degree, provides a fresh perspective. However, if this nanoscale perspective is not well-utilized for an in-depth understanding, then little science would be gained from the current study. Importantly, the origin of such kinetic stabilization is still not revealed, and many of the experimental designs are not scientifically sound or even conflict with each other. To name a few:

(1) There is a significant discrimination between major control samples (Fig.1c vs. Fig.1d). They have very different thermal histories at high T, and, understandably, they show different dispersions. Instead of controlling the total time (~1000 s or only 10 pulses), the authors should do 10,000-20,000 pulses (500-1000s at high temperature) for a fair comparison. The so-called kinetic stability may not be valid because of the kinetic advantages, but may largely be from a much shorter duration at high temperature. This can sabotage the foundation of this paper.

Our response: We thank the reviewer for raising this important concern. We fully agree that a fair comparison between pulsed heating and conventional heating must consider the total time the nanoparticles (NPs) spend at high temperature. Below we clarify our experimental design and provide additional experiments that directly address this point.

First, the comparison shown in **Figure 1c** and **Figure 1d** is initially designed to isolate the roles of heating/cooling rate in pulsed heating and conventional heating by ensuring that both methods share the same holding time at 1000 °C (50 ms). Although this comparison highlights the effect of rapid thermal cycling, we understand that it does not fully address the reviewer's request for an equal total high-temperature exposure.

Following the reviewer's suggestion, we thereby compare conventional heating (~1200 s total time under high-T) with 100 heating pulses (~1300 s cumulative high-T

exposure), where the total durations represent the sum of heating/cooling and isothermal holding periods. As shown in **Figure 3** and **Figure S17**, Pt NPs subjected to 100 heating pulses retain markedly better size stability and dispersion than those treated by conventional heating (**Figure 2e-h** and **Figure S7**) under this time-normalized condition.

To determine whether the observed stabilization arises from the intrinsic kinetic pathway of pulsed heating rather than simply from a shorter high-T duration, we conduct a new control experiment in which we significantly increase both peak duration and pulse number. Specifically, the sample was held at 1000 °C for 1 s per pulse and subjected to 1000 heating pulses (i.e., 1000 s of cumulative exposure at 1000 °C). As shown in **Figure R1** (also refer to the updated Supplementary Information as **Figure S19**), the evolutions for number and average size of Pt NPs during the first 100 pulses follow same trends as those in **Figure 3**, and critically, they remain stable without observable sintering even after 1000 pulses. This finding directly demonstrates that pulsed heating stabilizes NPs through a kinetically constrained pathway: the ultrafast heating/cooling rate and short high-T residence within each pulse limit long-range atomic diffusion and coalescence, while still allowing local structural relaxation and interface strengthening. By contrast, under conventional heating, the NPs are held well above the Tamman temperature (namely, diffusion of materials becomes “significant” above an empirical value of about half of the melting temperature)¹ of Pt (~750 °C) for long periods, leading to bond breaking and therefore NP migration and coalescence.

In this revision, we have added the above result to the revised manuscript together with the following text: “To demonstrate that pulsed heating stabilizes NPs through a kinetically constrained pathway rather than a short high-temperature exposure, we perform an extended pulsed heating of 1000 pulses with a 1 s peak duration (cumulative exposure of 1000 s at 1000 °C). Again, Pt NPs remain stable without observable sintering (**Figure S19**), and the evolutions for number and average size of Pt NPs during the first 100 pulses follow similar trends as those in **Figure 3**.”

Reference:

(1) Golunski, S. E. Why use platinum in catalytic converters? *Platinum Met. Rev.* **51**, 162 (2007).

Figure R1. (Figure S19) (a) In situ STEM image series showing the evolution of Pt NPs on graphene support during 1000 heating pulses, with a heating/cooling rate of $150\text{ }^{\circ}\text{C s}^{-1}$ and a peak temperature duration at $1000\text{ }^{\circ}\text{C}$ for 1 s. (b) Number and average size

of Pt NPs as functions of pulse number for (a). Error bars represent the 95% confidence interval of the mean.

(2) In the last round, the authors were asked to test the isothermal stability of Pt/graphene. They responded that this material can sustain 1000 °C for 1 hour without sintering (reply 4, Fig. S18). This result conflicts directly with Fig. 5g-h where nanoparticles show sintering starting at 6 seconds.

The reviewer tends to believe that the latter is the true case, as sintering is much easier at 1000 °C, where nanoparticles want to sinter and the graphene substrate is graphitized. As long as you do not have strong Pt-C/O bonds, the sintering is almost unavoidable. In the paper, the authors discussed having some crystalline affinity of Pt and the graphene, but no direct bond formation. Then the Pt clusters can not be stabilized at such a high temperature. This will be contrary to common sense.

Fig. 5g-5h showing sintering in 6 seconds at 1000 °C:

Reply and Fig. S18 showing anti-sintering in 1 hour at 1000 °C:

Our response: We thank the reviewer for this comment and for highlighting the need to clarify the apparent discrepancy between the isothermal test (**Figure S18**) and the sintering behavior observed in **Figure 5g-h**.

First, we would like to clarify that the two experiments as mentioned by the reviewer probe two distinct structural states of Pt NPs, which induces different thermal responses. **Figure 5g-h** captures the behavior immediately after the first pulsed-heating step to 1000 °C, during which the Pt precursor has only just decomposed into nascent clusters (i.e., “fresh” NPs). At this early stage, the metal-support interaction is undeveloped, and the NPs have not yet undergone any structural or interfacial relaxation. As a result, when these freshly formed clusters are held at 1000 °C for several seconds, substantial sintering occurs (as expected for weakly bound Pt on a pristine graphene surface). In contrast, **Figure S18** examines the isothermal stability of the fully evolved Pt/graphene nanocatalyst produced after 100 heating pulses (i.e., “annealed” NPs). Over the course

of these repeated pulses, the Pt NPs gradually undergo crystallographic refinement, facet selection, and interfacial reorganization (**Figures 3k, 4, and 5d**), and they develop much stronger interactions with the graphene support. Our EELS analysis (**Figure 2n**) together with our previous work² demonstrates interfacial charge transfer as well as formation of strong Pt–C interfacial bonds. In short, the stable behavior observed in **Figure S18** DOESN'T conflict with **Figure 5g-h**; rather, the two experiments highlight two different stages of NP evolution: an initially unstable state vs. a kinetically optimized, strongly anchored state.

To avoid potential ambiguity, we have revised the manuscript to clarify this point. The relevant sentence now reads as: “To further verify the kinetic stability of Pt NPs, we track their sintering behavior during the first pulse by rapidly heating them to 1000 °C and then holding for a duration (**Figure 5g**). At this early stage, the interaction between the NPs and the support has not yet developed.”.

Second, the stability of Pt/graphene nanocatalysts produced by pulsed heating is attributed to two mechanistic factors. (i) The structural adjustment during 100 heating pulses yields NPs with improved crystallinity and orientations aligned with the graphene support, producing lower-energy, more stable configurations (**Figures 3k, 4, and 5d**). (ii) The strengthened interfacial interaction (evidenced by C K-edge shifts in EELS in **Figure 2n**) is consistent with our previous findings, which indicates substantial charge transfer and the formation of strong Pt-C bonding at the interface.² These synergistic effects collectively generate the thermally robust nanostructures observed in the 1 h isothermal test (**Figure S18**). This conclusion is also consistent with reports showing that pulsed-heated nanocatalysts exhibit superior stability during extended thermal or reactive operation, attributed to metastable structures and/or enhanced metal-support coupling.³⁻⁵

Finally, we should explain that the graphene support does not undergo notable graphitization under our pulsed-heating conditions. The C K-edge spectra after 100 and 1000 pulses (**Figures R2 and R3**) show no detectable change; this agrees with the well-

established high thermal stability of graphene as a catalyst support.^{6,7} Hence, the observed stabilization of Pt NPs arises from the NP-support interface evolution rather than the chemical transformation of support.

Under this circumstance, we have inserted additional discussion in the updated text, which read as “The marked sintering resistance can be attributed to the formation of strong Pt-C interfacial bonding and the progressive structural alignment of NPs with graphene substrate, both of which arise during the prior heating pulses.”.

References:

- (2) Gan, M. et al. Incomplete charge transfer bestows significant sintering resistance for metal nanoparticles on two-dimensional graphyne. *J. Mater. Chem. A* **12**, 29174-29183 (2024).
- (3) Yao, Y. et al. High temperature shockwave stabilized single atoms. *Nat. Nanotechnol.* **14**, 851-857 (2019).
- (4) Li, C. et al. Ultrafast self-heating synthesis of robust heterogeneous nanocarbides for high current density hydrogen evolution reaction. *Nat. Commun.* **13**, 3338 (2022).
- (5) Du, P. et al. In-situ Joule-heating drives rapid and on-demand catalytic VOCs removal with ultralow energy consumption. *Nano Energy* **102**, 107725 (2022).
- (6) Jang, I. et al. Instantaneous thermal energy for swift synthesis of single-atom catalysts for unparalleled performance in metal-air batteries and fuel cells. *Adv. Mater.* **36**, 2403273 (2024).
- (7) Qiao, Y. et al. 3D-printed graphene oxide framework with thermal shock synthesized nanoparticles for Li-CO₂ batteries. *Adv. Funct. Mater.* **28**, 1805899 (2018).

Figure R2. (Figure S13) (a, b) STEM image and C K-edge EELS spectra of graphene support before and after 100 heating pulses.

Figure R3. (a, b) STEM image and C K-edge EELS spectra of graphene support before and after 1000 heating pulses.

(3) This question is not answered at all in the last round.--- What is the safe region for ultrafast heating to maintain particle stability, for example, T range, t or pulse times range, and heating/cooling ranges? The authors should dig into the mechanism of this kinetic stabilization and propose design principles, instead of just listing one example.

We understand the current pulse enables stability under 10 pulses. What we don't know is where the kinetic boundary is and the reason behind this boundary, for example, the kinetic competition between interparticle diffusion and the pulsed heating freezing. The author also mentioned heating rate is particularly important, but is this supported by experimental proof (must with fair controls)? Quantitative or numeric study will be largely preferred to understand this kinetic stability or its safe boundaries.

Our response: We sincerely thank the reviewer for this important and insightful question. Following your suggestion, we have explicitly mapped out the kinetic boundary (“kinetic window”) within which pulsed heating suppresses sintering in this revision (using the Pt/graphene system as a model).

We provide quantitative boundaries that define the conditions under which sintering is effectively suppressed. These boundaries emerge from the interplay between bond-

breaking/diffusion and the limited time NPs spend above the Tamman temperature during each ultrafast pulse. Our new experiments address all three key parameters requested by the reviewer (i.e., heating/cooling rate, pulse duration, and peak temperature) and establish clear mechanistic design principles.

1) The heating/cooling rate sets the onset of the kinetic regime. Prior studies typically define ultrafast heating as rate $\geq 100 \text{ }^\circ\text{C s}^{-1}$,⁸ and our side-by-side in situ TEM experiments confirm this threshold. At $100 \text{ }^\circ\text{C s}^{-1}$, the evolution of Pt NPs over 100 pulses mirrors the stabilization behavior observed at $150 \text{ }^\circ\text{C s}^{-1}$ (**Figure R4**), whereas further reducing the rate to $50 \text{ }^\circ\text{C s}^{-1}$ results in continuous coarsening with no sign of convergence in particle number or size (**Figure R5**). This arises because the slower heating rate prolongs the dwell above the Tamman temperature of Pt ($\sim 750 \text{ }^\circ\text{C}$), resulting in long-range diffusion. Thus, heating/cooling rate $\geq 100 \text{ }^\circ\text{C s}^{-1}$ is required to restrict the time available for bond breaking and migration of NPs.

2) The allowable dwell time at high temperature imposes a complementary kinetic constraint. While earlier single-pulse tests suggested that holding at $1000 \text{ }^\circ\text{C}$ for $<6 \text{ s}$ mitigates sintering (**Figure 5g**), our systematic study across multiple durations (50 ms, 2 s, 3 s, 4 s, 5 s, and 6 s, each repeated sequentially for 10 pulses) reveals a sharper boundary. As shown in **Figure R6**, pulse durations $\leq 4 \text{ s}$ well maintain the stabilization behavior, whereas the durations of 5–6 s trigger rapid coalescence (**Figure R6**). These results establish $\sim 4 \text{ s}$ as the time boundary for pulsed heating at $1000 \text{ }^\circ\text{C}$.

3) As the aim of pulsed heating is to produce well-crystallized NPs with an appropriate size distribution, the temperature window itself is bounded by the precursor decomposition threshold ($\sim 400 \text{ }^\circ\text{C}$) and the melting point of Pt NPs ($\sim 1005 \text{ }^\circ\text{C}$ for 2 nm particles).⁹ Within this range, higher temperatures increase the thermodynamic driving force for sintering, and our experiments show that even near the melting point ($1000 \text{ }^\circ\text{C}$), sintering remains effectively suppressed as long as the rate and duration remain within the kinetic window (**Figure R4**). This ensures complete precursor decomposition, rapid crystallization, and access to kinetically controlled structural relaxation pathways.

4) We also note that the number of pulses does not dictate kinetic stability. Even after 1000 pulses, corresponding to a cumulative 1000 s at 1000 °C, Pt NPs remain stable (**Figure R1**). Additional pulses primarily refine particle crystallinity and strengthen the metal–support interface, with these processes largely completed within ~100 pulses.

In summary, our supplementary quantitative analysis shows that pulsed heating suppresses sintering when three criteria are met simultaneously: (i) heating/cooling rate ≥ 100 °C s⁻¹ to minimize the effective diffusion time, (ii) high-temperature dwell time ≤ 4 s to prevent the onset of long-range coalescence, and (iii) peak temperature between precursor decomposition and melting point of NPs. These experimentally defined limits establish a clear kinetic mechanism and a practical design framework for applying pulsed or otherwise temporally modulated thermal stimuli to stabilize supported nanostructures.

References:

- (8) Chen, Q. et al. Rapid synthesis of metastable materials for electrocatalysis. *Chem. Soc. Rev.* **54**, 4567-4616 (2025)
- (9) Wang, G., Xu, Y.-S., Qian, P. & Su, Y.-J. The effects of size and shape on the structural and thermal stability of platinum nanoparticles. *Comput. Mater. Sci.* **169**, 109090 (2019).

Figure R4. (a-j) In situ STEM image series showing the evolution for Pt NPs on graphene support during 100 heating pulses, with a heating/cooling rate of 100 °C s⁻¹. (k) Number and average size of Pt NPs as functions of the pulse number for (a-j). Error bars represent the 95% confidence interval of the mean. The evolution of NPs over 100 heating pulses aligns with the previous observations at rate of 150 °C s⁻¹ (**Figure 3a-h**), wherein the NPs gradually stabilize on the support, indicating that the rate of 100 °C s⁻¹ remains within the kinetic window of sintering resistance. Combined with the results at a heating/cooling rate of 50 °C s⁻¹ shown in **Figure R5**, we therefore conclude that the kinetic boundary for heating/cooling rates is ≥ 100 °C s⁻¹.

Figure R5. (a-j) In situ STEM image series showing the evolution for Pt NPs on graphene support during 100 heating pulses, with a heating/cooling rate of $50\text{ }^{\circ}\text{C s}^{-1}$. (k) Number and average size of Pt NPs as functions of the pulse number for (a-j). Error bars represent the 95% confidence interval of the mean. The NPs exhibit continuous sintering throughout the 100 heating pulses, indicating that this rate is far from the kinetic window.

Figure R6. (a) In situ STEM image series showing the evolution for Pt NPs on graphene support during 60 heating pulses, with a varying duration of 50 ms, 2 s, 3 s, 4 s, 5 s, and 6 s for each duration within 10 pulses. (b) Number and average size of Pt NPs as functions of the pulse number for (a). Error bars represent the 95% confidence interval of the mean. The number and size of NPs gradually stabilize during the heating pulses with durations ≤ 4 s; however, as the duration increase to ≥ 5 s, sintering recommences and becomes more pronounced, indicating that the kinetic stability has been disrupted. These results confirm that the safe range for pulse duration is ≤ 4 s.

Point-by-point response

We sincerely thank the reviewers for their comments and suggestions on our manuscript. We have addressed all the concerns raised by the referees in full and revised the manuscript accordingly. **Our responses** to the reviewers' comments are given below.

The content is organized as following:

Section I: Response for Reviewer 1 (p2)

Section II: Response for Reviewer 2 (p3-p8)

Response for Reviewer #1

This study played a crucial role in elucidating the mechanism for rational nanostructured catalyst design through heat treatment in a metastable state. Furthermore, I believe it is fully acceptable for accurately answering the questions posed.

Our response: We sincerely appreciate the reviewer's positive assessment and recommendation for publication.

Response for Reviewer #2

The authors have made a substantial effort to address the concerns raised in the previous round by providing more in situ TEM evidence. Overall, the revised manuscript is largely improved in rigor and scope, and it convincingly demonstrates that pulsed heating can yield supported Pt nanoparticles with markedly enhanced resistance to sintering. However, one key concept and (mis)understanding should be clarified.

1. It remains insufficiently clarified on the roles of pulsed heating (PH) and Pt–C interfacial bonding in governing the long-term stability of the nanoparticles. In comment #2, the authors showed that, under PH operation, Pt NPs are not stable for the initial stage because their crystallinity and interfacial bond are not yet formed; The Pt NPs then become stable after ~100 PH cycles, even when annealed under isothermal heating at 1000 °C (does this mean isothermal heating can also freeze sintering?).

This experimental data indicate that the stability or long-term anti-sintering of Pt NPs is NOT (at least not entirely) from pulsed heating effect, but rather, and importantly, from the strong interaction with the substrate. This conclusion is understandable and convincing, but it is in strong contradiction to the authors' claim that pulsed heating freezes sintering in supported metal nanoparticles.

A clearer separation of the early-stage kinetic suppression of sintering and the late-stage chemical or interfacial stabilization, along with a discussion of the exact role of pulsed heating, would substantially strengthen the mechanistic clarity of the work.

It seems that, compared with slow continuous heating, PH is effective in building strong metal-support interaction at the initial stage by competing with sintering and graphitization behaviors. But, the long-term stability of nanoparticles under PH is not because of PH freezing, but because of the formation of strong interfacial bonding. The authors should either prove experimentally or theoretically that pulsed heating can indeed freeze sintering, or attribute the true role of PH to first forming the strong

bonding, and then ensuring stability.

Overall, the discussion of beyond thermodynamics, pulsed heating freezes sintering, the thermodynamically unstable yet kinetically stable state under PH, etc, lacks scientific rigor and experimental/theoretical proof. The authors should elaborate the paper strictly based on their observations.

Another one, why are responses and data in comment #3 not revealed in either the main text or SI? The discussion of possible kinetic competition among crystallization, bond formation, sintering, and graphitization is helpful for readers to understand the role and design of PH.

Our response: We thank the reviewer for this thoughtful and technically important comment. We fully agree with the reviewer's constructive comment on the role of pulsed heating in suppressing sintering. Actually, we have clearly stated that pulsed heating is currently adopted in the synthesis of supported nanocatalysts in the manuscript, and we are thereby working on the underlying mechanisms for these synthesis efforts in this work (which is "the early-stage" as raised by the reviewer).

Besides, in response to the reviewer's question regarding previous Comment #3, we have now included the full datasets (**Figs. R1-R3**) defining the kinetic window (heating/cooling rate, pulse duration, and peak temperature) in the Supplementary Information (**Supplementary Figs. 21-23**) and explicitly integrated the corresponding discussion into the revised manuscript and Supplementary Information, which reads as

"Consequently, our work offers clear mechanistic insights that NP stabilization for a specific metal during pulsed heating can be achieved within a kinetic window (which should be jointly governed by heating/cooling rate (**Supplementary Figs. 21 and 22**), pulse duration (**Supplementary Fig. 23**), and peak temperature, as discussed in **Supplementary Discussion**);" and "Using the Pt/graphene system as an example, we define the kinetic boundaries in which pulsed heating suppresses NP sintering. In situ STEM experiments show that heating/cooling rates ≥ 100 °C s⁻¹ stabilize Pt NPs over

100 pulses, whereas slower rates allow coarsening (**Supplementary Figs. 21 and 22**). Pulse durations ≤ 4 s at 1000 °C preserve progressive convergence of particle size and number, while longer pulses trigger rapid coalescence (**Supplementary Fig. 23**). Besides, the temperature window lies between the precursor decomposition (~ 400 °C) and the size-dependent Pt melting point (~ 1005 °C), with sintering effectively suppressed even near 1000 °C if rate and duration criteria are met (**Supplementary Fig. 21**). Pulse number does not affect stability, although additional pulses further improve crystallinity and metal-support bonding (**Supplementary Fig. 19**). These results establish a clear kinetic mechanism and provide design rules for NP stabilization via pulsed or temporally modulated heating: rates ≥ 100 °C s⁻¹, dwell ≤ 4 s, and temperatures between decomposition and melting.”.

Fig. R1. (Supplementary Fig. 21) (a-j) In situ STEM image series showing the evolution for Pt NPs on graphene support during 100 heating pulses, with a heating/cooling rate of 100 °C s⁻¹. (k) Number and average size of Pt NPs as functions of the pulse number for (a-j). Error bars represent the 95% confidence interval of the mean.

Fig. R2. (Supplementary Fig. 22) (a-j) In situ STEM image series showing the evolution for Pt NPs on graphene support during 100 heating pulses, with a heating/cooling rate of $50\text{ }^{\circ}\text{C s}^{-1}$. (k) Number and average size of Pt NPs as functions of the pulse number for (a-j). Error bars represent the 95% confidence interval of the mean.

Fig. R3. (Supplementary Fig. 23) (a) In situ STEM image series showing the evolution for Pt NPs on graphene support during 60 heating pulses, with a varying duration of 50 ms, 2 s, 3 s, 4 s, 5 s, and 6 s for each duration within 10 pulses. (b) Number and average size of Pt NPs as functions of the pulse number for (a). Error bars represent the 95% confidence interval of the mean.